# From short to long term: Dynamic analysis of FDI and net export in global regions

**Sanduni Lakshani**[1], **Chanuka Sandaruwan**[1], **Chanaka Fernando**[1], **Gayan Vidyapathirana**[1], **Ruwan Jayathilaka**[2]*, **Sumudu Munasinghe**[1]

**1** Sri Lanka Institute of Information Technology, SLIIT Business School, Malabe, Sri Lanka, **2** Department of Information Management, Sri Lanka Institute of Information Technology, SLIIT Business School, Malabe, Sri Lanka

* ruwan.j@sliit.lk

**Data Availability Statement:** All relevant data are within the manuscript and its with supporting information files.

## Abstract

It is crucial to examine the impact between foreign direct investment (FDI) and net exports (NE) for unveiling international trade dynamics, and the economic development of different geographical regions. It yields sharp insights into how FDI inflows, driven by theories such as backward linkage, export platform, and knowledge transfer, enhance a host country's export capacity and contribute to economic growth. Moreover, studying the reciprocal linkages between FDI and NE helps recognise the aspects of domestic factors, such as productivity and the product life cycle, in attracting FDI and increasing export performance. Based on those theories, the study aims to ascertain the dynamic causality or correlation between FDI and NE across all regions with the utilisation of panel data gathered from 110 countries, considering the period from 2002 to 2020. The Wavelet coherence method is used to investigate the relationship between these variables across different frequencies and periods, followed by a Granger causality test. The findings demonstrated that FDI and NE have a significant relationship in most regions, with a bidirectional relationship between FDI and NE across all continents. The results could assist respective governments and policymakers in formulating policies related to FDI flows and offer insights into how a host country can attract more FDI and boost NE.

## Introduction

International trade and foreign direct investment (FDI) are often linked and contribute to a country's economic growth in multiple ways [1,2]. Moreover, the dynamic relationship between FDI and trade in developing and developed countries is of significant theoretical and empirical interest among scholars [1,3–9]. Furthermore, the role of both exports and FDI in economic growth has been highlighted in recent literature.

Global FDI has become a significant phenomenon in recent years due to globalisation, the rapid expansion of international trade, economic interdependence, and inter-regional trade agreements [10]. Moreover, international trade is essential because no economy in the world is entirely self-sufficient. While the export led growth hypothesis postulates that exports are the primarily driver of overall economic development [11], the empirical evidence shows that FDI

**Funding:** The authors received no specific funding for this work.

**Competing interests:** The authors have declared that no competing interests exist.

flows have been expanding at a rate that outpaces the growth in the volume of international trade [12]. However, there is still disagreement and different opinions among the export-led growth literature and the FDI growth literature.

Several theoretical and empirical studies confirm that there is a complementary relationship between FDI and trade, highlighting how FDI inflows positively influence a country's net exports vice-versa [1,3–8]. However, some scholars argued that FDI and trade react as substitutes in the presence of international trade barriers [9,13]. This means that there is a negative relationship between FDI and trade. In addition, there are also some empirical studies that have failed to find any significant relationship between FDI and trade [14]. This phenomenon occurs when the returns from foreign investment, such as facilitating access to international markets, enable the transfer of expertise and new technology, increase local capital, and develop local human capital through training and the promotion of export industries in the host country [15]. Hence, the policy decisions relevant to international trade and foreign investment are crucial in determining the future direction of a country.

As the world over the years, FDI inflows have steadily and substantially increased from US$ 751.69 billion in 2002 to US$ 1.28 trillion in 2020 [16]. The impact of FDI inflows on a host country has become a subject of ongoing discussions in recent years. Meanwhile, NE have also substantially increased and reached US$ 22.48 trillion in 2020 [17]. However, when considered region-wise, a striking feature of the last two decades is that the regional behaviour of FDI and NE differs and is not consistent with global trends (Fig 1).

Although many scholars have already widely discussed this comprehensive relationship based on different geographical regions, previous studies were most likely based on a single or a few geographical areas, and still, there is no study found as continent-wise analysis in past literature. Moreover, there are still different opinions and disagreements in past literature, and different results have been shown in different borders and regions.

A study based on African regions discovers a close link between FDI and NE period for 1980 to 2007, confirming the positive one-way effect of FDI on the balance of trade through export promotion [8]. However, another study offered a completely different perspective on the relationship between FDI and NE based on 8 developing countries in Asia, and the study found that there is a bidirectional linkage between exports and economic growth in the short run, and exports have a long-run effect on FDI from 1986 to 2013 [2]. In another recent empirical study that conducted in European Union countries, it was revealed that FDI investments increase trade flows in the countries being considered, with no signs of a relationship between the variables [19].

Some empirical studies in the American region have revealed that the impact of FDI inflows on Latin America's trade balance is positive with a one-way relationship from 1970 to 1994 [20–22]. These findings were validated by another study conducted in Mexican states, revealing that high FDI impacts the export of products in border areas in Mexican states from 2007 to 2015 [23]. Accordingly, it appears that the relationship between FDI and trade cannot be generalised for every region, and the nature of the relationship may change depending on regional cooperation, growth, rules and regulations, policies, and other factors.

On the other hand, the study based on the Organization for Economic Cooperation and Development (OECD) states that the temperament and behaviour of the relationship also change over time, reflecting the complexity of the FDI and trade nexus [24]. The study further revealed that international trade helped to promote more FDI until the middle of the 1980s. However, the direction of the relationship between FDI and international trade changed after this period [24].

According to the past literature, there has been no study covering all regions to compare the results of each region and specific country, and there is a need for a study at the regional

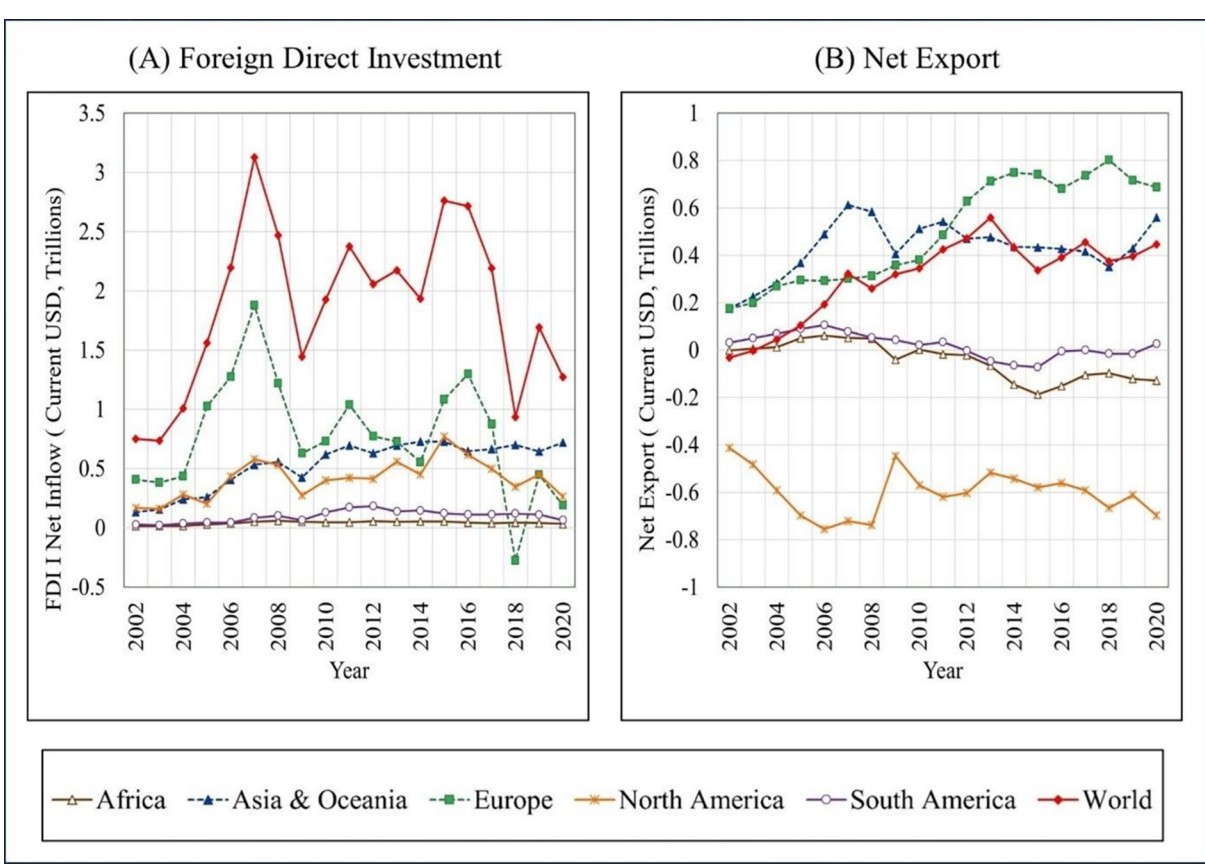

**Fig 1. Region-wise behaviour of FDI inflows and net export, 2002–2020.** Source: Authors' Compilation based on reported World Bank data [18].

level. Therefore, this study provides a detailed explanation of the causality and correlation of FDI on NE across a broad scope at different times, in contrast to prior studies based on a single country or small geographic area.

Accordingly, the study's main objective is exploring the linkage between NE and FDI inflows, considering variation in different periods over 19 years. This study was undertaken as a continent-wide study to investigate variations across multiple regions concurrently. Furthermore, to enhance comprehension of the relationship between NE and FDI at the country level, a Granger causality test was conducted for each individual country using the annual time series data period for 2002–2020. Accordingly, the study contributes to the advancement of the body of literature in three ways. Firstly, the novel utilisation of the Wavelet coherence approach adds a new perspective to exploring the nature of the relationship and causality between NE and FDI. This method can capture short and medium-term changes in the direction of the relationship during the specified study period, which previous studies could not discern. In addition, even slight changes at long-term, short-term, and medium-term time scales are reflected in these results, which should have been addressed in previous studies. Secondly, the study covers a global perspective, considering over 110 countries from all habitable continents and spanning nearly two decades. Existing literature has focused on specific regions or individual countries, and no comprehensive global study compares all habitable continents. Finally, by conducting a country-wise analysis to examine the behaviour of individual countries to

gain further insight into the relationship between NE and FDI, this study contributes a comparative approach to the existing literature.

Moreover, a Granger causality analysis was also carried out to understand further the relationship between NE and FDI in each country. The test results will aid in accurately identifying the effect and its direction between the variables. Hence, a significant justification is provided along with these results by utilising two analysis methods.

The article begins with a review of the findings of previous scholars who have conducted studies on a similar focus while an explanation of the data and methodology used is discussed next. The latter part. Will present the results and discussion, with an overall conclusion of the study.

## Literature review and theoretical framework

### Theoretical framework

The linkage between NE and FDI is based on theoretical motives that help explain their interactions. These theories provide frameworks for confirming the dynamics between NE and FDI and reveal their underlying mechanisms.

**How FDI affects net export.**   According to the backward linkage theory, FDI creates backward relationships with local suppliers and expands their export capacities. The impact of FDI on backward linkages in Indonesia was examined and found that FDI inflows lead to increased demand for domestic inputs, which in turn support export-oriented industries and stimulate NE [25].

Knowledge and technology transfer theory suggests that FDI helps to transfer knowledge, technology, and managerial competence from investing companies to the host country. This transfer uplifts the productivity and competitiveness of local companies, enabling them to manufacture higher-quality products and services for export [26]. An empirical study revealed the effect of FDI on knowledge transfer and export performance in developing countries. The results indicated that inflows of FDI were positively linked with technological spillovers and improvements in export capabilities, thereby facilitating the idea that knowledge and technology transfer through FDI can enhance NE [26].

Productivity theory argues that FDI prompts productivity growth in the host country, guiding a positive impact on NE. By proposing new technologies, manufacturing techniques, and management practices, FDI uplifts the efficiency and productivity of local industries. A study investigated the linkage between FDI, productivity, and NE in China, revealing that inflows of FDI led to enhanced productivity levels in local firms, which in turn resulted in higher NE [27].

**How net export affects on FDI.**   The export-platform theory demonstrates that NE can captivate FDI as companies seek to use the host country as a platform for exporting to alternative markets. A Mexican study examined the linkage between FDI and exports, which supports the theoretical backing. The evidence explored that FDI inflows guided to enhanced exports, particularly in industries with higher foreign ownership, supporting the notion that net exports can play like a magnet for FDI by serving as an export platform [28].

Product life cycle theory posits that FDI is influenced by the life cycle of products. When a product matures and demand increases in international markets, firms may invest in foreign countries through FDI to set up manufacturing facilities near the target markets. In an empirical study, the linkage between product life cycles, FDI, and NE was analysed. The study revealed that as products transitioned from the introduction stage to the maturity stage, FDI enhanced, leading to a shift in net export patterns as production became more internationally dispersed [29].

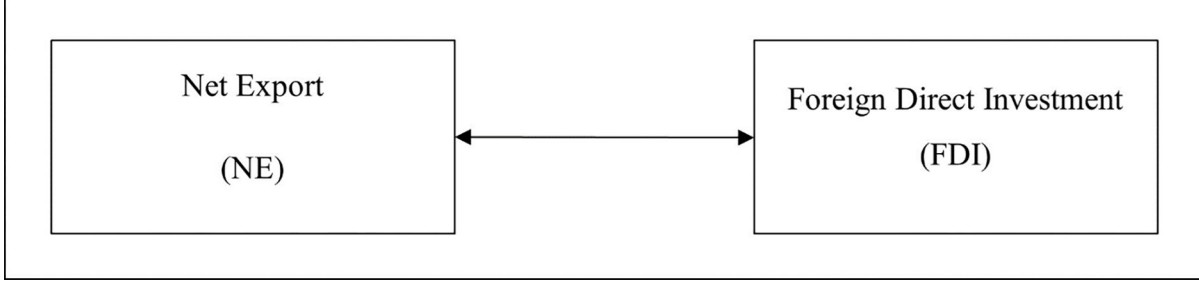

**Fig 2. Theoretical framework interrelationship of variables.** Source: Authors' demonstration based on literature.

Internalisation theory suggests that companies invest in foreign countries through FDI to internalise their business operations. NE can influence the decision to internalise business activities via FDI as companies' goal is to secure their existence in foreign markets and have better control over their export operations. An empirical study investigated the effect of internalisation pros on the choices of the locations of multinational enterprises. The empirical findings demonstrated that internalisation advantages like access to foreign markets played an important role in companies' decisions to attract FDI [30].

According to the theoretical evidence, the theoretical framework (Fig 2) was constructed to investigate the relationship between FDI and NE.

Prior empirical studies support the theoretical framework and all factors considered in this study. The theoretical framework clearly depicts the study aim, which is to identify/understand the relationship between NE and FDI. Thus, the study presents the hypothesis to examine the objective. Furthermore, strong theoretical support is evident in the hypothesis of regarding the relationship between FDI and NE. As a result, the study develops a hypothesis for examining the correlation and/or causality between NE and FDI across all regions and different time frequencies.

## Literature review

Considering the determinants of FDI, NE can be considered a key factor attracting FDI. This section highlights the importance of the dynamic correlation between these variables for all regions at the global level. Based on various prior literature evaluations, the effects of the FDI on NE and the reverse are discussed. Fig 3 shows the filtering process performed for the literature review.

**European region.**   A study examining FDI, exports, and growth transition for ten European countries indicated that FDI leads to exports only in Poland. Latvia shows a bidirectional relationship between FDI and exports [31]. The study considered data from 1994 to 2008 and applied the techniques of the Autoregressive distributed lag bound test (ARDL BT) approach and Granger causality test. However, the scope was limited to ten countries in the European region, and quarterly data has been considered in the analysis to illustrate the precise relationship between FDI and NE. In a case study based on Turkey, analysed the impact of FDI on exports was analysed using the ARDL BT approach and Cointegration analysis techniques [7]. There, the effect of FDI on exports was identified as statistically significant, and these tests confirmed a positive relationship between FDI on exports.

Field examined the function of FDI on the growth of international competitiveness in a study that collected data from the Czech Republic, Hungary, Poland, and Slovakia [1]. The results were estimated using regression models from 1996 to 2018, and there was an indication of a positive effect of FDI inflow on the export growth of all the countries of the Visegrad

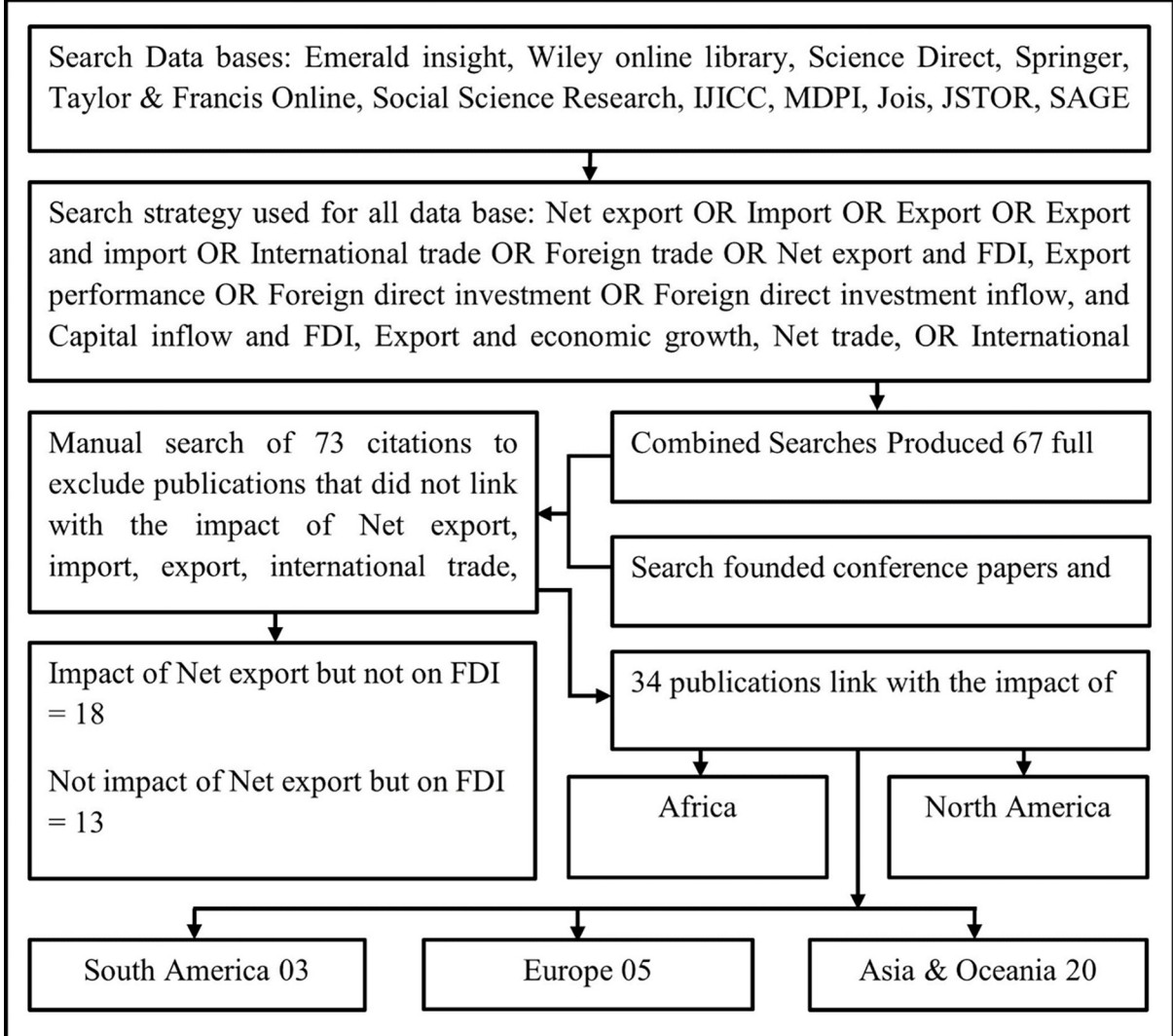

**Fig 3. Literature search flow diagram.** Source: Based on authors' observations.

Group. Moreover, researchers investigated the impact of direct investment inflows on macro-economic variables such as imports and exports, from 1974 to 2014 in Turkey [32]. This study revealed a unidirectional relationship between FDI to exports and FDI to imports using the Augmented Dickey–Fuller test (ADF) Unit Root, Johansen cointegration, Vector Autoregressive (VAR) lag, and Granger causality tests models. The overall results indicate that FDI has a positive relationship with Turkey's trade.

On the other hand, in a recent empirical study investigated the FDI flows of three groups of countries from the European Union (EU), namely the Visegrad Group and the Euro area, from 2005 to 2019 on a comparative basis [19]. The EU included Romania and Bulgaria, and the Visegrad group included Poland, Hungary, Czech Republic, Slovakia and 19 countries in the Euro area. The findings revealed that FDI investments increase trade flows in the countries being considered, but there were no indications of a relationship between the variables. However, the authors suggested considering several regions in future studies to understand the influence between the variables.

**African region.** Focusing on studies conducted in the African region from 1980 to 2007, researchers presented empirical evidence to conclude that the expansion of FDI in the African region positively affects export promotion and trade balance [8]. The study used the Least square dummy variable (LSDV) regression method to investigate the relationship between FDI and trade balance (export and import), revealing significant elasticity in exports and imports. At the same time, imports are more elastic than exports. This suggests that Multinational enterprises (MNEs) in Sub-Saharan Africa (SSA) are not only dependent on exports but also imports to a greater extent.

Exports of 13 SSA countries between 1970 to 2017 reveal the relationship between FDI and economic growth. The Granger causality test was also used to obtain the variables in pairs based on the Bayesian estimation of the Seemingly Unrelated Regression (SUR) system. The results concluded that Benin, Mauritania, Nigeria, and South Africa show a one-way relationship from FDI to export, while Burkina Faso, Madagascar, Malawi, Nigeria, and Rwanda offer a one-way connection from export to FDI [33].

In Ethiopia, considering the period from 1922 to 2018, a study was conducted to study the empirical relationship between FDI and export performance using the ARDL model. The findings indicated a statistically insignificant relationship between FDI and export performance in Ethiopia [34]. Even though this study, which has investigated the FDI and export performance only in the context of Ethiopia in the African continent, covered ninety-six-year years, which has yet to be focused on in any current research. However, there is a lack of research on studies in the African region in the literature covering multiple countries, with an increased need to conduct extensive research with a broader scope creating more room for generalisation.

In another study, annual data from 1970 to 2010, was used to explore the impact of key macroeconomic variables (economic growth, exports, gross capital formation, trade openness, inflation) on FDI in Nigeria [35]. The ARDL technique was used for this study and the Wavelet coherence technique may be used to further confirm the results. Additionally, Fully modified ordinary least square (FMOLS) and Dynamic ordinary least square (DOLS) were utilised to test the robustness of ARDL long-run estimation. According to financial statements from the ARDL long-run estimate, exports and trade openness have a positive effect on FDI data. This study examined data collected over several years longitudinally to gain a better understanding of the relationship between FDI and macroeconomic variables. As a result, a clear relationship between the factors can be revealed. In addition, to the best of the authors' knowledge, no previous research has used Wavelet coherence and Wavelet correlation approaches to investigate these dynamics.

Moreover, conducted this study to examine the long-run relationship between FDI and exports using the ARDL model considering the period from 1980 to 2015 [36]. The findings show that FDI in Nigeria has a positive and statistically significant effect on total exports. Further, it appears that FDI in primary sectors and manufacturing sectors has a positive and significant long-run relationship with both total exports and oil exports, but FDI in the services sector has no significant effect on Nigerian exports.

**North American region.** Considerable focus was given to studying the causal link between FDI and NE in the North American region. In Mexico, the economy, FDI inflows and manufacturing exports were identified as crucial factors affecting the economy. This study investigated the impact of FDI on manufacturing export performance among Mexican states and regions from 2007 to 2015 using static and dynamic panel data states, revealing that different Mexican states respond differently to FDI fluctuations in the short term [23]. It has been confirmed that high FDI impacts the export of products in border areas. Additionally, internal and external factors affecting manufacturing exports have been investigated in the study with the application of the Generalised Method of the Moments (GMM) technique. However, this

study could have indicated a more precise picture of the relationship between FDI and NE in this region. Moreover, when it comes to studies on exports, it is considered more appropriate to conduct extensive research on various sectors rather than being limited to one industry. Further, due to the lesser amount of research conducted to demonstrate the relationship between exports and FDI in the North American continent, it is necessary to conduct more new and current research.

**South American region.** In the case study of Brazil, it was examined that with the increase in exports, the inflow of FDI has also increased, and FDI is affecting exports [22]. The study aimed to understand whether there is evidence that an increase in FDI benefited Brazil's trade balance from increased imports and exports from 1995 to 2007 and whether there is a relationship between FDI and the trade balance. A Granger causality test was used to verify the results, and finally, the results concluded that there was an increase in exports and that FDI affects imports in the short run but not in the long run, and exports in both time frames.

Some studies argue whether increased FDI has boosted Brazil's foreign trade in the long run and whether there is a stable relationship between a firm's FDI strategy and international trade [21]. Furthermore, generalised linear models and Moderated multiple regression (MMR) methods were used to examine how FDI affected the export and import of 11 sectors in Brazil from 1996 to 2009. The evidence confirmed that FDI was associated with increased exports in the short run and revealed a positive relationship between FDI and exports in the long run due to export-intensive industries with export priority. In addition, it is concluded that among import-oriented sectors, a positive relationship between FDI and imports has been shown in the short run, and a negative relationship between FDI and imports has been shown in the long run. However, both studies mentioned above are based only on Brazil in the South American region. Therefore, since not many countries are in the area, conducting research based on the entire region appears more appropriate. And since there is a limited amount of research based on this region, there is a need to develop the focus in future studies.

Latin American and Southeast Asian countries from 1970 to 1994 revealed the relationship between foreign trade and FDI [20]. The results have shown that imports affect FDI in Latin America, Argentina, Ecuador, and Peru, and FDI affects exports in Chile. Finally, considering both regions, it was concluded that the impact of FDI inflows on Latin America's trade balance is positive.

**Asian & oceanic region.** Several scholars have focused on studying the effects between FDI and NE in Asia & Oceania. For instance, a study on China's exports by foreign affiliates interacting with FDI reached $81 billion in 1998 [37]. This study attempted to illustrate the relationship between China's exports and inward FDI promotion using panel data collected at the provincial level between 1986 and 1997. The results confirmed that higher FDI levels positively affect provincial manufacturing export performance. Similarly, another study examined the impact of FDI on the export performance of China's provinces and highlighted a positive and significant indication of the relationship between FDI and exports across all Chinese regions from 1984 to 1997 [4]. Additionally, a study investigated the relationship between inward FDI and exports in China as a whole during the period 1986 to 1999 [38]. The study found a bidirectional relationship between FDI and China's regional export trade using synchronisation/Error correction modelling (ECM). However, all three studies mentioned above were limited to China alone, and since the data was collected between 1980 and 2000, there is a need for more up-to-date and comprehensive studies covering a broader scope beyond China.

Another piece of Chinese empirical evidence reveals the relationship between FDI inflows and bilateral trade from 1980 to 2003. Granger causality has been used to achieve the results of this study, and based on those results [5] it was concluded that FDI is positively related to exports and imports in the long run and the short-term by using the Vector error correction

model (VECM) framework, there is a bidirectional positive relationship between FDI and exports. In addition, it has been revealed that there is a unidirectional relationship between import to FDI. Export performance can be introduced as an essential factor in determining the competitiveness of a country's industries and the rapid development of a country [3]. This study has primarily analysed based on India. It has investigated the impact of FDI inflows into India. In addition, the effects of FDI inflows in India on export performance have also been explored. This study is based on the period from 1991 to 2007 and uses regression analysis; its empirical results concluded that FDI inflow and India's export performance have a significant positive relationship. This study has investigated the relationship between FDI and NE through a simple regression analysis based only on India in the Asian region. However, since there are many countries in the Asian region, it is more appropriate to investigate the effect of FDI on the overall export performance of most countries by not confining it to a single country and using diverse analytical techniques.

India waited four decades to introduce a new import substitution policy and finally launched a new economic model in 1991 to deal with the financial crisis [39]. This study investigated the patterns of FDI in India from 1991 to 2011. At the same time, a VECM has been used to infer the relationship between FDI and manufactured exports based on annual time series data analysis. According to this study, the overall results conclude that there is a bidirectional causality between FDI and exports. A study has been conducted to investigate the dimensions of Vietnam's international trade and investment and the impact of FDI on export growth by comparing Thailand, the Philippines, Malaysia, Singapore, and Indonesia after Vietnam's economy transferred into an open market-oriented economy, and the Inflow of FDI for Vietnam was around USD 36.8 billion in 2009 [40].

Another study has found a strong positive relationship between exports and FDI using the fixed effects model between 1970 and 2004 for 49 countries based on the World Development Index (WDI) 2005 [41]. Similarly, a study was conducted to examine the correlation between FDI, exports and economic growth in eight developing countries in Asia from 1986 to 2013 using the VECM framework [2]. Presenting an entirely different perspective, the study revealed that exporting has a long-term impact on FDI. In addition, it is concluded that there is a bidirectional relationship between exports and economic growth in the short run. But since this study was conducted only for eight selected developing countries in the Asia region, it is impossible to generalise these findings to the entire area or a much broader context. Also, a long run relationship between FDI and export was evident, with no apparent specific effect in between. Contrary to the findings of the above study, a Jordanian study confirmed that FDI inflows strongly affect international trade [42].

Further detailing the study, the impact of FDI on exports from 1980 to 2018 was determined by adopting ARDL BT and cointegration approaches [42]. Here, the data revealed a positive relationship between FDI and long-run exports in Jordan. However, this study was limited to more than one country in Asia, although it was investigated between an extended and long-term time interval, so the researchers could not see a more comprehensive picture of a region. Furthermore, a study has been conducted to examine how FDI affects agricultural exports in Arab countries [43]. This study was born from 2000 to 2019 using panel data from thirteen Arab countries. This study used Lagrange Multiplier (LM) test followed by methods such as Pooled Regression Model (PRM), Fixed Effect Model (FEM) and Random Effect Model (REM). These results concluded that FDI has a positive relationship with agricultural exports. However, this study focused on investigating the effect of FDI on agricultural exports in Arab countries only. However, it is viewed as essential for scholars to have a look at total exports to get a better understanding of the relationship between FDI and exports.

A similar conclusion is that although the impact of FDI on economic growth, trade, domestic investment and other areas of the economy has been investigated, how FDI affects Export processing zones (EPZs) has not been explored [44]. Accordingly, the effect of FDI inflow on EPZs in Bangladesh in terms of employment and exports has been investigated using secondary data from 1997 to 2018. Again, models such as GMM and ARDL have been used to achieve the results. Finally, the results concluded that FDI positively affects exports in the EPZs and that the inflow of FDI between employment in the EPZs positively impacts the long and short term. A positive relationship is found between exports and FDI, providing further evidence of these variables [6].

ARDL cointegration is used to analyse the impact of FDI on trade in Sri Lanka for time series data from 1980 to 2016 [45]. The findings of this study, through short-run and long-run estimations demonstrated a substantial positive linkage between the two variables. Additionally, the study emphasises that ARDL and Granger causality test are used to investigate the impact of FDI on India's export performance [46]. The results show that though ARDL confirms a negative effect of FDI on exports in the short run, there is a unidirectional relationship between FDI and exports through Granger causality with the use of data from 1980 to 2017. Using the same methodology, a bidirectional relationship between FDI and exports is also revealed, providing further evidence over the same period [47].

A recent study aimed to investigate the relationship between Bangladesh FDI inflows and export performance, taking into account structural breaks, using annual time series data from 1972 to 2019 [48]. Using the VECM model, the authors found a positive unidirectional effect of real exports (REX) on real FDI (RFDI) in Bangladesh. The study further emphasised the use of structural break methods to investigate this matter. Another Bangladeshi empirical evidence reveals the relationship between FDI inflows and export performance in the long run as well as in the short run from 1995 to 2020. Johansen cointegration and VECM methods were used to obtain the results, and the findings confirm that there is a significant statistical relationship between FDI inflows and export performance in the long run [49]. Similarly, in Jordan, scholars investigated how foreign trade (FT), inflation rate (INFR), gross domestic product (GDP), interest rate (IR), and FDI interact with macroeconomic variables using commonly used approaches such as unit root, cointegration, and ARDL [50]. The final data revealed a statistically significant positive linkage between FDI and international trade in Jordan. The authors also discovered that GDP and international FT have a statistically significant positive link, whereas INFR and exports have a statistically significant negative relationship.

In a global context, a dynamic relationship between FDI, corruption and economic growth in 54 developed and developing countries between 1996 and 2018 was investigated using a Panel vector autoregression (PVAR) model [51]. Trade openness, total credit to the private sector (CPS), and exchange rate volatility (EV) are among the other control variables utilised in the analysis. The findings reveal that corruption control has a negative (positive) effect on foreign investment and economic development in developing (developed) countries, implying that weaker (stronger) institutional quality and more (lower) corruption increase investment and economic progress. According to the study, economic growth and corruption have a positive bidirectional relationship in developing countries but a negative unidirectional relationship in developed countries. In addition, the authors further find that trade openness (exports and imports of goods and services) is positively associated with FDI inflows.

Reviewing the results of these past studies, it is evident that various studies focus on the effect of FDI on exports, with the absence of a comprehensive analysis covering all continents. Moreover, only a few studies were conducted to analyse the causal relationship between FDI and export using Granger causality and Wavelet coherence methods. However, scholars have many disagreements regarding the nature of the influence between these variables. Therefore,

this study contributes to filling this gap by providing an in-depth comparative analysis of the relationship between FDI and NE at the global level.

## Data and methodology

### Data

In this study, panel-time series data were used to examine the variables of NE and FDI, covering the period from 2002 to 2020. The data were collected from the World Bank, considering this specific timeframe due to various significant factors that could impact the relationship and variation in FDI and NE. Notably, the global financial crisis and the global oil crisis of 2007–2008 were key factors influencing FDI and NE fluctuations during 2002 to 2022 [52]. Additionally, the years 2019 and 2020 were critical due to significant events impacting the global economy, such as Russia's invasion of Ukraine and the spread of the COVID-19 pandemic [52]. Thus, the period from 2002 to 2022 was chosen to conduct a comparative investigation of the effects of these events on the correlation between NE and FDI.

Although data from the World Bank sources were available starting from 1960, it was not uniformly available for all countries. Therefore, data from 110 countries were included in the analysis, covering 19 years. The original data were used for the study, and to facilitate the analysis, all figures were converted into USD millions. The study encompasses 34 European countries, 28 Asian and Oceania countries, 21 African countries, 17 North American countries, and 10 South American countries. It is important to note that the authors included three (3) countries from the Oceania region within the Asia & Oceania region category.

Under the United Nations (UN) classification, all countries worldwide are categorised into developed and developing nations [53]. Among the selected countries 72 countries are classified as developing nations, while 38 are considered developed nations. For a comprehensive overview of the countries and continents included in the study, please refer to S1 Appendix. The secondary data sources used for the study are presented in Table 1 and the data file used for the analysis is provided in S2 Appendix.

### Methodology

**Wavelet analysis.** The Wavelet coherence technique was employed to assess causality and/or correlation in time series data [54]. Initially developed in 1984 [55], the Wavelet coherence approach has been extensively discussed in various studies [35,56–59] and has found applications in diverse fields, including medicine [60], finance [61,62], economics [63], and tourism [64].

Empirical studies have highlighted significant advantages of the Wavelet coherence technique over other methodologies. For example, in the analysis of the linkage between stock market returns and macroeconomic variables, Wavelet coherence analysis outperformed traditional techniques by providing a better understanding of frequency-dependent relationships [65]. Another study investigating the connection between land surface temperature and

**Table 1. Data sources and definition of variables.**

| Variable | Definition | Measure | Source |
|----------|-----------|---------|--------|
| FDI | Foreign direct investment | Data are in current United States Dollars (USD) | The World Bank (16) |
| | | | https://data.worldbank.org/indicator/BX.KLT.DINV.CD.WD |
| NE | Net export | Data are in current United States Dollars (USD) | The World Bank (17) |
| | | | https://data.worldbank.org/indicator/BN.GSR.GNFS.CD |

Source: Authors' compilation based on The World Bank (18).

landscape patterns demonstrated the superiority of Wavelet coherence analysis in capturing time-varying and scale-specific coherence [66]. These findings underscore the significance of Wavelet cohere analysis in revealing complex and dynamic relationships. Additionally, the techniques was applied to examine inter-annual and inter-decadal oscillations in monthly precipitation extremes and their teleconnections to climate indices, revealing significant coherence at specific frequency bands that conventional techniques failed to capture [67].

Real-world data often exhibits subtle fluctuations that can be crucial for gaining insights. While Fourier analysis can express certain trends, it fails to capture sudden changes effectively. A Wavelet is a rapidly decaying wave with zero mean and limited duration, available in various shapes and sizes [68]. There are two main type of Wavelet transforms: Continuous and discrete which differ based on the scaling and drawing of these waves.

The significance of Wavelet coherence analysis lies in several key aspects. Unlike other linear and nonlinear analysis methods, it does not require data pre-treatment, enabling the rapid filling of data for multiple time intervals. This aids in generating visual representations of the output instead of mere numerical values. Moreover, the analysis considers the data along short, medium, and long-term dimensions, visually presenting complex and extensive data in a simplified manner. This capability allows for a better understanding of the overall patterns and relationships in the data.

A Wavelet, which was built by the function $\psi^{a,b}(x)$ ($\psi$ denotes Morlet Wavelet function), with constructions, $a$ and translation, $b$ can be represented as follows,

$$\psi^{a,b}(x) = |a|^{-\frac{1}{2}}\psi\left(\frac{x-b}{a}\right) \tag{1}$$

Wavelet coherence can be considered as an augmentation of the Wavelet analysis. Wavelet coherence explicit how one variable leads to another in bivariate analysis. Wavelet coherence is superior to other analysis such as Granger causality because of the ability that can explain intensity and direction of the relationship between two variables in different time period. With the investigation of two time series, Wavelet coherence is ablet to explain the direction from one variable to another. The Wavelet coherence formula [63], including smoothing factor, s, first variable, y and second variable x, is denoted as follows,

$$Coherence = \frac{|sWave.xy|^2}{sPower.x \cdot sPower.y} \tag{2}$$

The study considered NE as the first variable, and the second variable was the FDI. The data set that used in the study is attached in S2 appendix and Wavelet coherence graphs were illustrated using R software.

**Granger causality.** There are different approaches to testing causality in panel data series. This study used cross-country VAR Granger analysis [62,63,69,70], to evaluate the causality between FDI and NE for each country undertaken in the study. Initially, the study ran the Augmented Dickey–Fuller test (ADF/dfuller) [71] and Phillips–Perron test (PP) [58] to assess the stationary ($H_0$: $\delta = 0$) and non-stationary ($H_A$: $\delta \neq 0$) of the two-time series [72]. Thus, ADF Eq is as below.

$$\Delta y_t = \alpha + \beta t + \gamma y_{t-1} + \delta_1 \Delta y_{t-1} + \cdots \delta_{p-1}\Delta y_{t-P+}\varepsilon_t, \tag{3}$$

ADF (dfuller) test results in a negative number; when the number is more negative, the hypothesis is rejected that there is a unit root with a high level of confidence. Here, $\alpha$ denotes

the constant of the time series. $\beta$ indicates the coefficient of the time series, and $\rho$ prescribes the lag order. Similarly, the PP test is derived from the dfuller test.

The Granger causality runs along each individual country to further enhancement of the result that obtained from Wavelet coherence technique. Here, $X$ and $Y$ are the two stationary covariance variables. The variable $X_{i't}$ causes $Y_{i,t}$, if we are better able to predict $Y_{i,t}$ by utilising all variable information, compared to the use of information except $X_{i,t}$ for each individual $i \epsilon [1, N]$. The test assumed that the model is linear, thus use time stationary VAR representation for a data set to each cross-section unit $i$ and time period $t$, considering those aspect the Eq 8 is estimated,

$$Y_{i,t} = \sum_{k=1}^{p} \beta_k Y_{i,t-k} + \sum_{k=0}^{p} \theta_k X_{i,t-k} + u_{i,t} \qquad (4)$$

Here, $u$ is normally distributed with $u_{i,t} = \alpha_i + \varepsilon_{i,t}$, $P$ represents the number of lags. The equation assumed that $\beta_k$ is the auto regressive coefficients and $\theta_k$ is the regression coefficients. Those coefficients are constant for $k \epsilon [1, N]$ [63].

## Empirical results and discussion

### Preliminary findings

An overview of the descriptive statistics of the main variables used in the study is presented in Table 2. Across all regions, a total of 2090 observations were reported. On average, the Net Exports (NE) for the regions analysed in this study were as follows: Globally US$ 1,959, Africa Region US$ 1,603, Asia and Oceania Region US$ 14,798, Europe Region US$ 12,179, South Region US$ 1,571, and North American region US$ -37,281, which is the lowest. Moreover, the average value of net FDI inflows for the regions of Global, Africa, Asia and Oceania, Europe, North America, and South America are US$ 13,859, US$ 1,182, US$ 14,797, US$ 18,730, US$ 22,066, and US$ 8,680, respectively. NE ranges from US$ -763,533 to US$ 358,572.60, with mean and standard deviation values of US$ 1,959.49 and US$ 66,728.22, respectively. FDI net inflows to the selected global sample between 2002 and 2020 ranged between US$ -344,707.70 and US$ 733,826.50, with a mean of US$ 13,858.98 and a standard deviation of US$ 46,851.42.

**Table 2. Descriptive statistics of the key variables.**

| FDI Inflow (Current USD) | Obs | Mean | Std. Dev. | Min | Max |
|---|---|---|---|---|---|
| Global | 2090 | 13,858.98 | 46,851.42 | -344,707.70 | 733,826.50 |
| Africa | 399 | 1,181.50 | 2,235.52 | -7,397.30 | 11,578.10 |
| Asia & Oceania | 532 | 14,797.24 | 37,859.53 | -25,093.14 | -136,063.39 |
| Europe | 646 | 18,730.13 | 57,412.20 | -344,707.70 | 733,826.50 |
| North America | 304 | 22,065.62 | 70,917.90 | -2,154.73 | 511,434.00 |
| South America | 209 | 8,679.93 | 17,874.63 | -1,904.35 | 102,427.20 |
| **Net Export (Current USD)** | | | | | |
| Global | 2090 | 1,959.49 | 66,728.22 | -763,533.00 | 358,572.60 |
| Africa | 399 | -1,602.91 | 7,354.63 | -34,677.10 | 26,025.09 |
| Asia & Oceania | 532 | 14,797.78 | 52,892.41 | 290,928.43 | 358,572.60 |
| Europe | 646 | 12,178.69 | 45,690.03 | -85,471.11 | 260,592.20 |
| North America | 304 | -37,280.61 | 138,374.60 | -763,533.00 | 45,678.11 |
| South America | 209 | 1,571.05 | 9,305.10 | -54,977.53 | 35,838.20 |

Note: Obs., SD, Min., and Max. represent Observations, Standard Deviation, Minimum value, and Maximum value, respectively.

Source: Authors' calculation based on data from the world bank.

Looking at the NE from 2002 to 2020, the continent of Asia and Oceania recorded the highest value of USD 358,572.60, and the lowest NE was recorded in the North American continent, with a value of -763,533 US$. The European region collected the highest value of net FDI inflows (US$ 733,826.50), as well as the lowest negative FDI, which was worth US$ -344,707.70. The North American region has the highest standard deviation of net FDI inflows, which is due to the high dispersion of FDI inflows in the region. Conversely, the lowest standard deviation is in the Africa region, as FDI inflows to all countries considered in this region are low, resulting in low dispersion and standard deviation. Additionally, North America and Africa have the highest and lowest standard deviations in terms of NE, respectively.

Fig 4(A) shows the average FDI in different regions at the global level over the period 2002 to 2020. The highest mean FDI inflows were recorded in 2007 from the European region. However, in 2009, a significant decrease in FDI was observed in the European region and this decline continued after 2019. The financial crisis, Russia's invasion of Ukraine and the spread of the Covid-19 pandemic during that time contributed to the decline in FDI in the European region [52]. Similarly, in 2017 and 2018, the North American region also experienced a sharp fall in mean FDI inflows. This was attributed to multinational firms in the United States repatriating accumulated gains after corporate income tax reforms were enacted in the aftermath of the global financial and oil crises [73].

The European continent showed the highest FDI value of US$ 48928.35 in 2007 and the lowest FDI value of US$ -5180.2 in 2018.The FDI trend in the world region during 2008 to 2009 indicates a slow trend after the global financial crisis. The financial crisis during 2008–2009 led to a decline in FDI due to increased borrowing by banks and investors, regulatory and policy failures [73]. In the North American region, FDI increased after 2014, reaching US$ 22,444.96 in 2014 and further increasing to US$ 38,833 by 2015. The World Bank attributed this increase in FDI to a decline in the fraction of the population living in poverty, which reached a new low in 2015 [74].

Fig 4(B) shows the region-wise mean of NE (2002–2020) in different regions at the global level over the period 2002 to 2020. The highest mean NE was recorded in the Asia and Oceania region in 2007 but significant drop occurred NE in 2018. However, the scenario stabilised afterward. The 2018 financial crisis affected both FDI and NE, with FDI impacting exports after 2018 due to the consequences of the Arab Spring and the global financial crisis, resulting in a minimum NE in the Asia and Oceania region [50]. North America showed negative NE every year from 2002 to 2020 compared to other continents, indicating that NE rises when FDI falls and vice versa. The strength of local currencies in the American region could be a contributing factor to this trend. On the other hand, the European region showed positive NE from 2002 to 2020, with the highest export in 2018. Turkey's strong trade sector contributed to the high NE in the European region [32].

The NE of the South American region showed a gradual positive fluctuation from 2002 to 2011, but it gradually declined from 2012 to 2015. In contrast, the African region recorded negative NE from 2008 to 2020. This is because Africa's exports are dominated by primary items with volatile prices and less competitiveness in the international market. Despite the significant growth of the country's economy in recent years, the contribution of exports to the overall GDP remains low [75].

## Empirical findings

An interpretation of Wavelet correlation graphs is provided to understand better the time and frequency correlation between FDI and NE in the short, medium, and long term. Table 3 represents the interpretation of Wavelet coherence analysis.

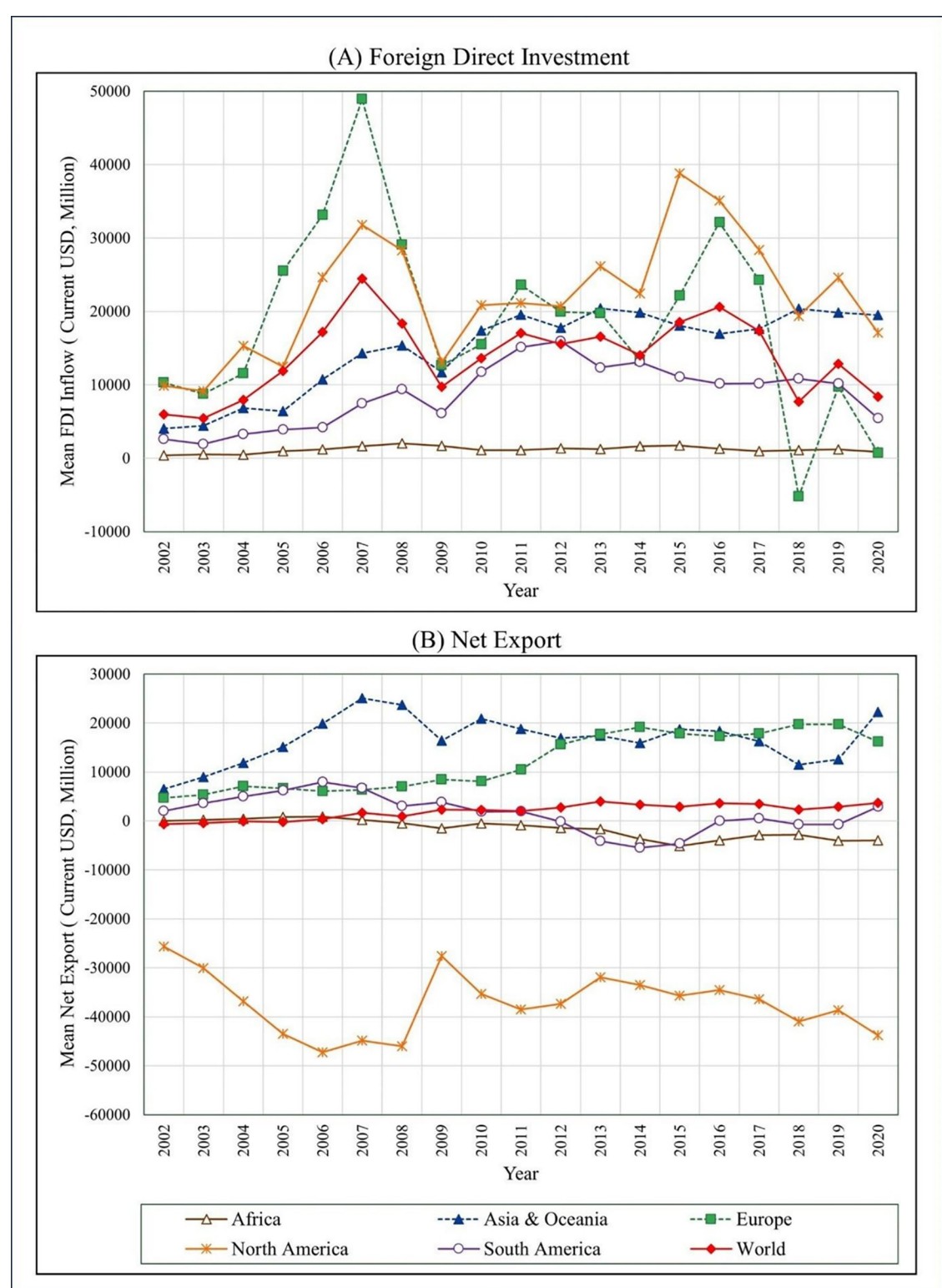

**Fig 4. Region-wise mean FDI inflows and Net Export, 2002–2020.** Source: Authors' Compilation based on World Bank data [18].

Table 3. Interpretation of Wavelet coherence.

| Direction of arrows | Interpretation |
| --- | --- |
| Rightward arrows | In-phase (Positive relationship) |
| Leftward arrows | Anti-phase (Negative relationship) |
| Rightward and up arrows | The FDI lead (cause) the NE rate |
| Rightward and down arrows | The NE lead (cause) the FDI |
| Leftward and up arrows | The FDI lead (cause) the NE |
| Leftward and down arrows | The NE (cause) the FDI |
| Cold (blue) | No correlation |
| Warm (red) | Correlation |
| Horizontal axis | Time period |
| Vertical axis | Scale |
| Low frequency | 0.0–0.3 |
| Medium frequency | 0.3–0.7 |
| High frequency | 0.7–1.0 |

Source: Authors' compilation.

The plot uses a horizontal axis to represent the time, with the leftmost area indicating the beginning of the stipulated period (2002) and the rightmost area indicating the end of the interval (2020). The vertical axis represents the period with lower frequency bands (higher frequencies) indicated at the top area of the plot, and higher bands (lower frequencies) represented at the bottom. The scale is divided into portions (upper, middle, and lower), indicating correlation for the short, medium, and long term, respectively. The blue (cold) region indicates no correlation, while the warm (red) region denotes correlated variables. The thick black line represents statistically significant areas of coherence with a 95% confidence level.

The arrows in the plot serve two purposes. Firstly, they indicate the correlation between the two-time variables at a specific point. The leftward arrows (←) denote an anti-phase relationship, indicating a negative correlation between the two variables. Conversely the rightward arrows (→) signify an in-phase relationship, indicating a positive correlation. Secondly, the direction of the arrows (upward and downward) shows which time series leads the relationship at that specific point. The downward arrows ($\nearrow$, $\searrow$) denote that the first time series (NE) leads the second time series (FDI), while the upward arrows ($\searrow$, $\nearrow$) demonstrate that the second time series (FDI) causes the first time series (NE). Figs 5–10 illustrate the causality and correlation between FDI and NE for the period 2002–2020.

In the third figure (Fig 5A), the right-facing arrows indicate a positive correlation between FDI and NE in the short-term (high frequency) of 2002, 2006, 2007, 2008, and 2012. Additionally, there is a mixed (positive and negative) relationship between FDI and NE during the brief period between 2017 and 2019. Left-facing arrows provide evidence of negative correlations in the short term in 2009 and 2014. In 2005, the high-frequency upward arrow to the right indicates that FDI leads to NE. On the contrary, in 2010, in the medium term, upward indicators to the left suggest that FDI leads to NE.

Moreover, from 2004 to 2009, the upper and lower arrows to the right with medium and low frequency indicate a positive relationship between FDI and NE. From 2016 to 2020 a cross-correlation between FDI and NE is observed with high, medium, and low frequency. Additionally, in 2013 and 2014, in the medium term, downward arrows to the right indicate that NE leads to FDI.

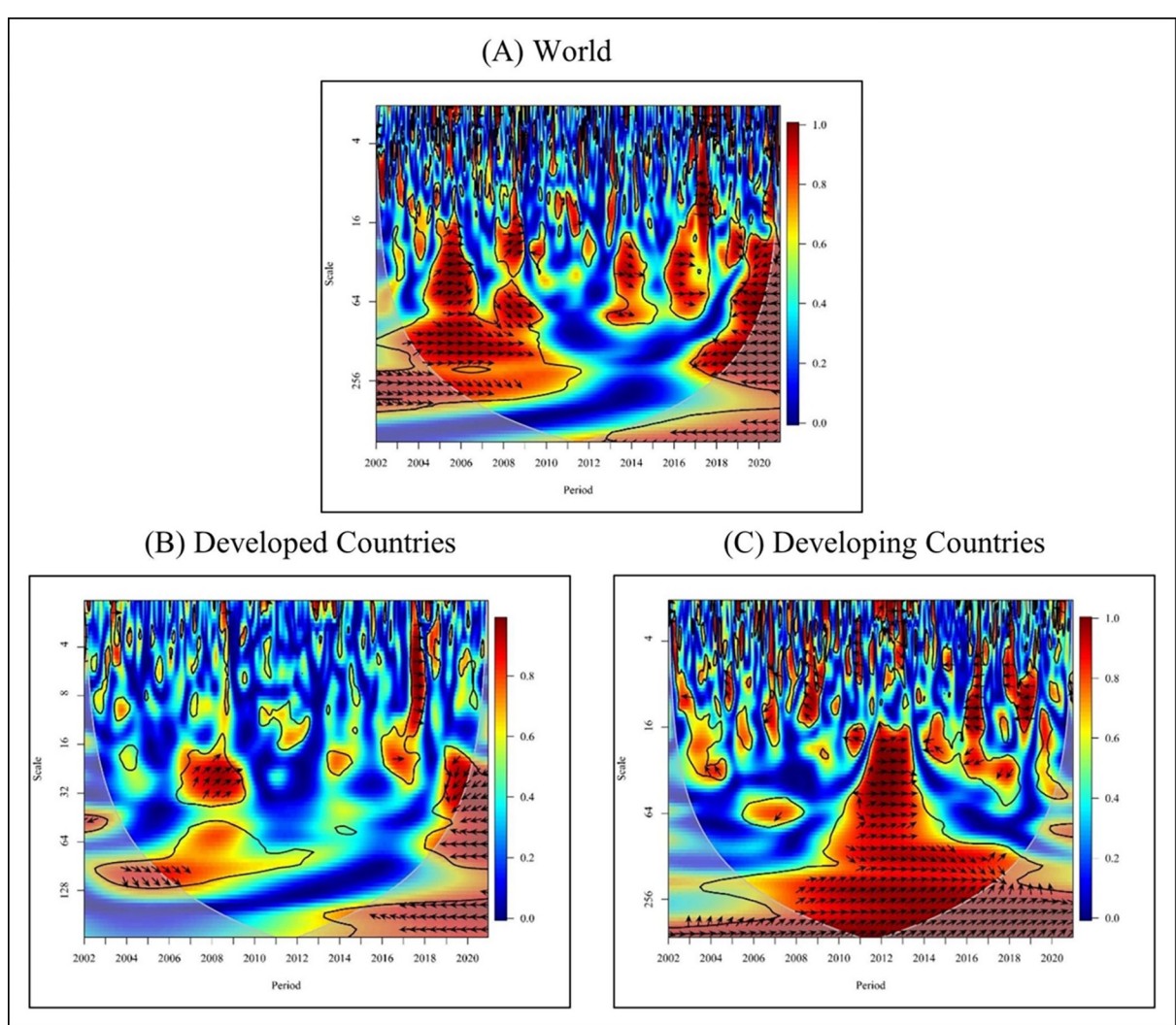

**Fig 5. Wavelet coherence: NE vs FDI for global.** Source: Authors' illustrations using R-Software.

Fig 5B, the graph shows the global relationship between FDI and NE in developed countries. In the medium term (medium frequency) from 2007 to 2009, the cluster of upward arrows to the right indicates that FDI leads to exports. These findings are consistent with those of who identified the effect of FDI on export growth in the Visegrad Group countries [1]. The findings support the notion that FDI positively affects the structure of international trade. Moreover, these results align with the findings of Basilgan who showed that FDI can boost exports in developing countries by providing better technology and capital [7]. Furthermore, the right-facing arrows indicate a positive relationship between FDI and NE in the short term (high and medium frequency) in 2004, 2009, 2017, and 2018. In the long run (low frequency) from 2005 to 2007, downward arrows to the right indicate that NE leads to FDI. Additionally, it is evident that there is a mixed relationship between FDI and NE with medium frequencies in 2019 and 2020. These findings are in line with the results of the study based on 54 developed and developing countries from 1996 to 2018 [51]. They emphasised that greater trade openness (exports and imports of goods and services) in developed countries lead to less trade barriers, and positive FDI inflow, sending positive signals to international investors. The study also confirmed positive relationship between trade openness and FDI inflow.

Overall, Fig 5C graphically depicts a mixed (positive and negative) correlation between FDI and NE in developing countries globally. However, from 2008 to 2018, the medium and long-term (high and low frequency) upward and downward arrows to the right indicate a positive relationship between FDI and NE. These findings align with the results noted by [4], who suggested that China's FDI inflows are growing due to increased long-term regional industrial competition in global markets. This increase in FDI has led to export growth and economic expansion in China, establishing a significant positive relationship between FDI and exports across Chinese provinces. Similarly, another study focusing on 13 Arab countries, found a positive impact of FDI on agricultural exports, with a 1% increase in FDI leading to a 0.055% increase in agricultural exports [43]. These findings further support the notion that exports increase when FDI increases and vice versa, confirming a significant positive relationship between FDI and agricultural exports.

Consequently, from 2014 to 2019, a negative relationship exists between FDI and NE with high and medium frequencies, as indicated by upward and downward arrows to the left. Additionally, the high and medium frequency arrows of 2003, 2004, 2005, 2009, 2011, 2015, and 2020 point upward to the left, indicating that FDI is directed towards NE. On the other hand, between 2012 and 2013, the upward and downward arrows to the right indicate a relationship between FDI and NE. With the implementation of India's liberalisation strategy, FDI inflows have increased, leading to a corresponding increase in exports. Consequently, India's ranking among other nations has risen with the inflow of FDI [3], and empirical evidence supports the positive effect on FDI on India's export performance. These findings are consistent with the results of other studies [51]. In summary, Fig 5C highlights the complex and dynamic relationship between FDI and NE in developing countries over time. The results show varying patterns of correlation and causality, which are supported by previous empirical studies and specific country cases. The arrows in the graph provide a clear visual representation of the temporal and directional aspects of this relationship, shedding light on the different impacts FDI can have on export performance in developing nations.

In Europe, from 2003 to 2004, the graph (Fig 6) shows negative causality by demonstrating leftward up arrows where the FDI leads to the NE in the short term (high and medium frequencies). However, FDI has a positive association with Turkish trade (imports and exports) from 1974 to 2017 [32]. Although this study finds a positive link during certain times, the above finding is negative, therefore the outcome is different. However, from 2006 to 2009, the plot depicted a rightward arrow uptrend by illustrating a positive correlation between FDI and NE. Empirical evidence showed that foreign investment inflows have a positive effect on export growth only in Visegrad Group countries [1]. Furthermore, FDI positively impacts the structure and dynamics of international trade. Visegrad Group Countries rapidly integrated into European business and investment networks after the collapse of state socialism. As a result, the exports of those countries grew. The result directly affects the positive effect between NE and FDI.

Nevertheless, from 2010 to 2013, the graph shows negative causality by demonstrating leftward down arrows where the NE leads to FDI in the short term (high frequencies). Furthermore, the arrows point down to the right from 2012 to 2013, where NE and FDI show a positive relationship with medium frequency. The positive impact of foreign direct investment on Turkey's import of goods and services can be attributed to the lack of raw materials, management, technology, and finance. Therefore, products are created using imported goods and services, and these finished products can then be exported to the world market [32]. Thus, this empirical study has confirmed the claims.

Overall, in the African region (Fig 7), the direction of the arrowhead is leftward. Arrows to the left indicate a negative relationship with the period from 2002 to 2007 except from 2004 to 2005, and FDI led to NE in the long term (High and Low frequency). From 2005 to 2007, the

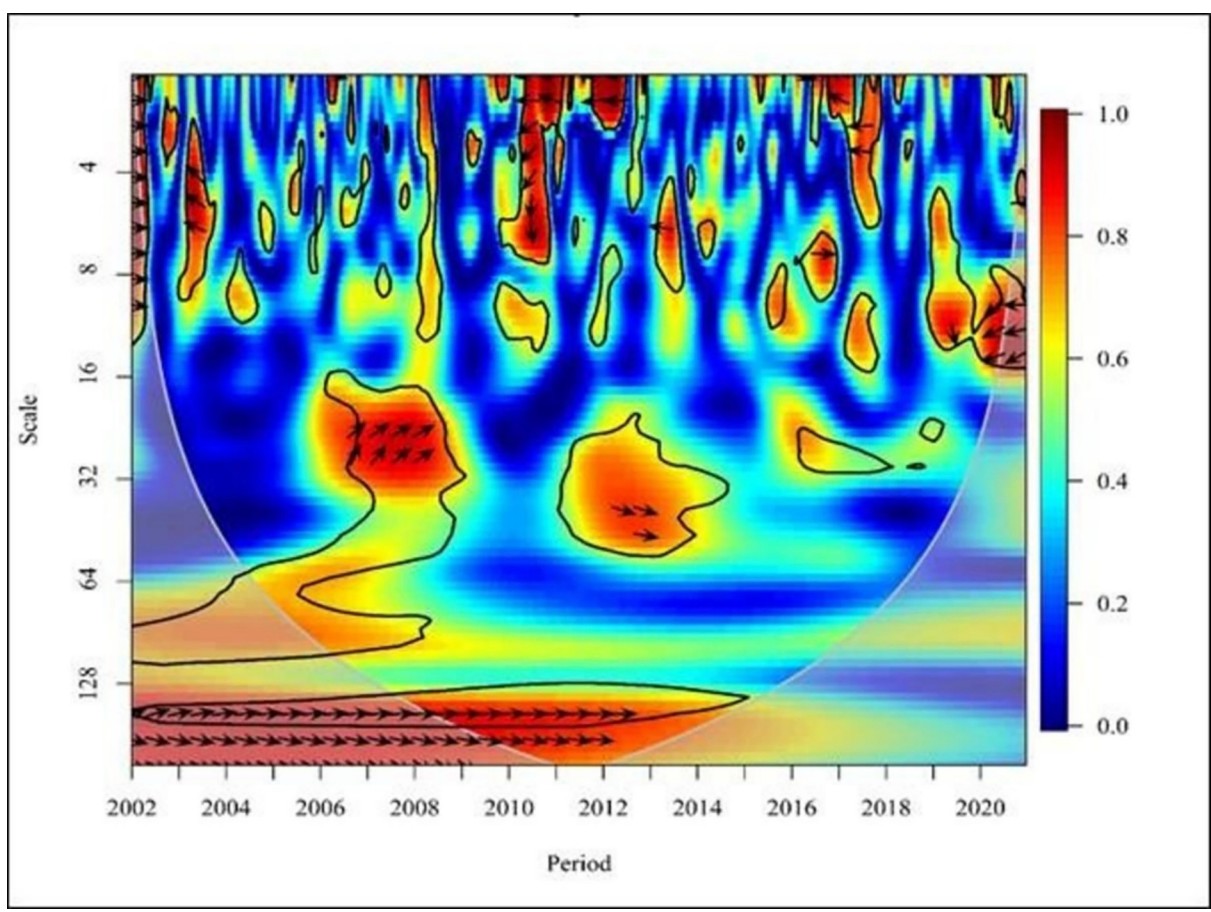

**Fig 6. Wavelet coherence: NE vs FDI for Europe continent.** Source: Authors' illustrations using R-Software.

graph shows negative causality in Africa by demonstrating leftward arrows where the FDI leads to the NE in the short term (High frequencies). Moreover, the arrows indicate a leftward and downtrend where the NE leads to FDI from 2007 to 2011 in the medium term (medium frequency). There is a negative relationship between NE and FDI. From 2016 to 2017, the graph shows negative causality by demonstrating leftward up arrows where the FDI leads to the NE in the short term and medium term (high frequencies). However, from 2015 to 2017, an arrow depicting a rightward uptrend illustrates a positive correlation between FDI and NE in the medium term (Medium and High frequency). These results support the findings of the study conducted by highlighting the impact of the two variables in a similar nature [8].

On the other hand, Africa's exports and imports have increased over the past few decades. A higher resilience of FDI as positive exports indicate that FDI significantly impacts Africa's export sector. When the FDI increases, the export will also increase; thus, the trade sector will also benefit. Many African nations benefit more from exports than imports because of the dwindling amount of their foreign exchange. In addition, the inflow of FDI is also growing in promoting MNEs based on exports. Due to this, the above factors prove that FDI positively affects exports as it can benefit from exports on the continent during the promotion of FDI. The FDI and NE have a negative relationship in Africa as a whole. It indicates that when FDI increases, NE decreases, and when FDI decreases, NE increases. However, most past literature evaluations have found a positive correlation between FDI and NE [8,33,35,36].

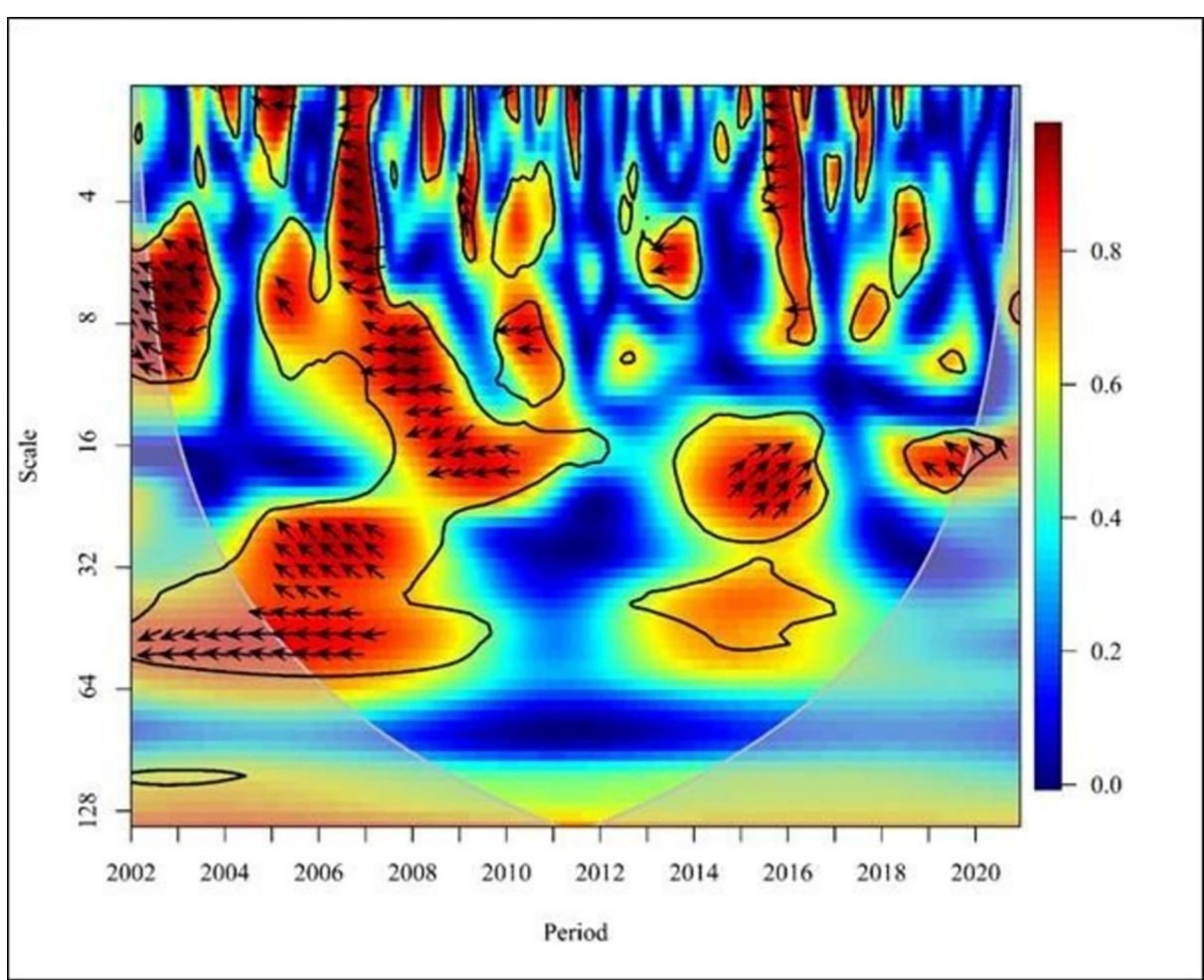

**Fig 7. Wavelet coherence: NE vs FDI for Africa continent.** Source: Authors' illustrations using R-Software.

In Fig 8, overall, the direction of the arrows in the North American region extends to the right and left. This predicts the mixed (positive and negative) effect between FDI and NE in the short-term, medium-term, and long-term. Arrows to the left indicate a correlation between FDI and NE in the short term (high and medium frequency) with an upward and downward trend from 2003 to 2019 except for 2002 and 2004 and 2005 to 2008, and 2014 and 2016. In addition, between FDI and NE from 2004 to 2019, an upward and downward arrow trend has been shown to the left with medium and long term (medium and low frequencies). On the contrary, the short-term down arrow to the right in 2005 shows a positive NE pointing to FDI.

Moreover, from 2004 to 2006 and from 2013 to 2015, the arrows show an upward and downward trend to the right between FDI and NE in the medium term. Furthermore, 2017 and 2018 have shown a cross-correlation between FDI and NE in the medium term. Overall, FDI negatively impacts NE in the North American region. In other words, NE increases when FDI decreases, or NE decreases when FDI increases. The reason for that can be pointed out to be the strength of the local currencies of the American region countries. It implies that countries with a robust domestic currency are more favourable for exports than imports [76]. In other words, a strong exchange rate discourages exports and encourages imports. Therefore, because of the strength of the

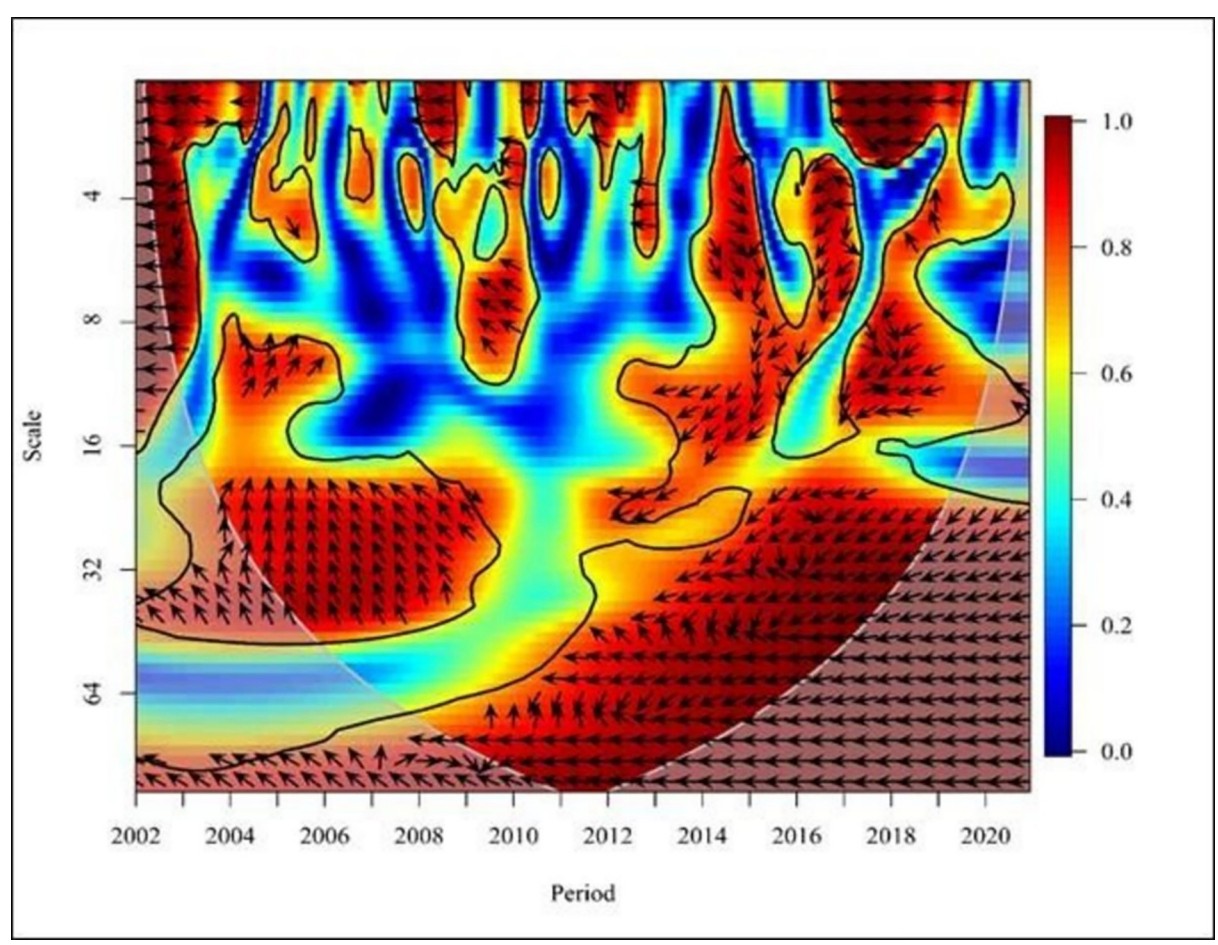

**Fig 8. Wavelet coherence: NE vs FDI for the North American continent.** Source: Authors' illustrations using R-Software.

domestic currency in the Americas, FDI in the Americas tends to be more import-oriented than export-oriented. Thus, due to the increase in FDI, imports and NE decrease.

In Fig 9, the left facing arrows provide evidence of a negative correlation between FDI and NE in the short term (high and medium frequency) from 2014 to 2016 in the South American region. Hence, the left and downward arrows indicate that NE leads to FDI. Furthermore, the arrows pointing up and down to the left represent a negative effect between FDI and NE with medium and low frequencies from 2005 to 2015. Additionally, in the short run between 2007 and 2008 and 2019, the upward arrows to the right indicate that FDI leads to NE. These findings are consistent with a study conducted in Brazil [22]. They revealed that there is an increase in exports in the short term as well as in the long term, while imports increase only in the short term. FDI also increases due to an increase in NE and a positive relationship between FDI and the trade balance has been shown. Exports are rising in Brazil's trade sectors, especially for market-related businesses. Hence, as exports rise, FDI inflows expand. For this reason, market growth will also increase, and thus economic expansion will also increase [20]. Thus, according to the above study, there is a positive relationship between FDI and NE in Brazil, consistent with the above findings. Here the FDI helped to increase the exports, and it has been confirmed that FDI affects exports. Accordingly, the effect of FDI inflows on Latin America's trade balance has been proven to be strong.

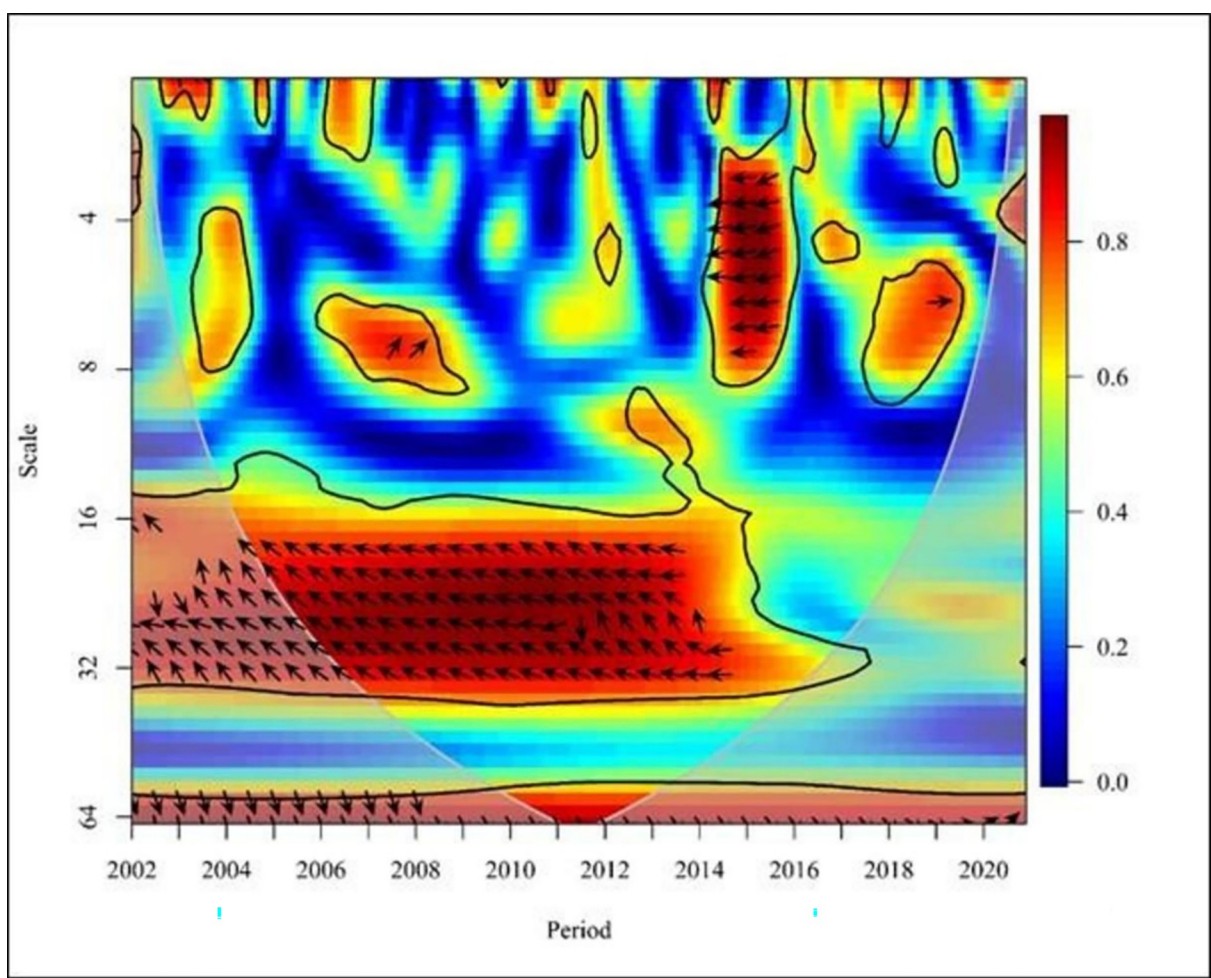

**Fig 9. Wavelet coherence: NE vs FDI for South America continent.** Source: Authors' illustrations using R-Software.

Fig 5A shows a positive correlation between FDI and NE in the Asia & Oceania region in the short term with high and medium frequency from 2003 to 2020, except from 2007 to 2010, 2013, 2015 and 2018. The results above demonstrate a positive linkage between FDI and NE between 2003 and 2012. These findings are consistent with the conclusions of the Wavelet method, providing evidence for a positive relationship between FDI and trade (exports and imports) in Sri Lanka [45]. Additionally, these findings are in line with the results of the study conducted in Jordan [50], which pointed out a positive relationship between FDI and foreign trade, with a 1% increase in FDI leading to a 13% increase in exports. This shows that an increase in FDI in Jordan is associated with an increase in exports, indicating that NE increases as FDI increases and NE decreases as FDI decreases. However, in 2015, short-term (high frequency) upward arrows to the left indicate that FDI leads to NE. The impact of FDI on India's export performance during the period 1980 to 2017 has shown a negative relationship using ARDL [46]. On the contrary, the same study shows a unidirectional relationship between FDI and NE using Granger causality. Accordingly, this study shows both similarities and dissimilarities, aligning with the result mentioned above. In 2019, the short-term downward indicators to the left suggest that NE leads to FDI. These findings are consistent with a study conducted in Bangladesh [48], which demonstrated a positive unidirectional impact of NE on

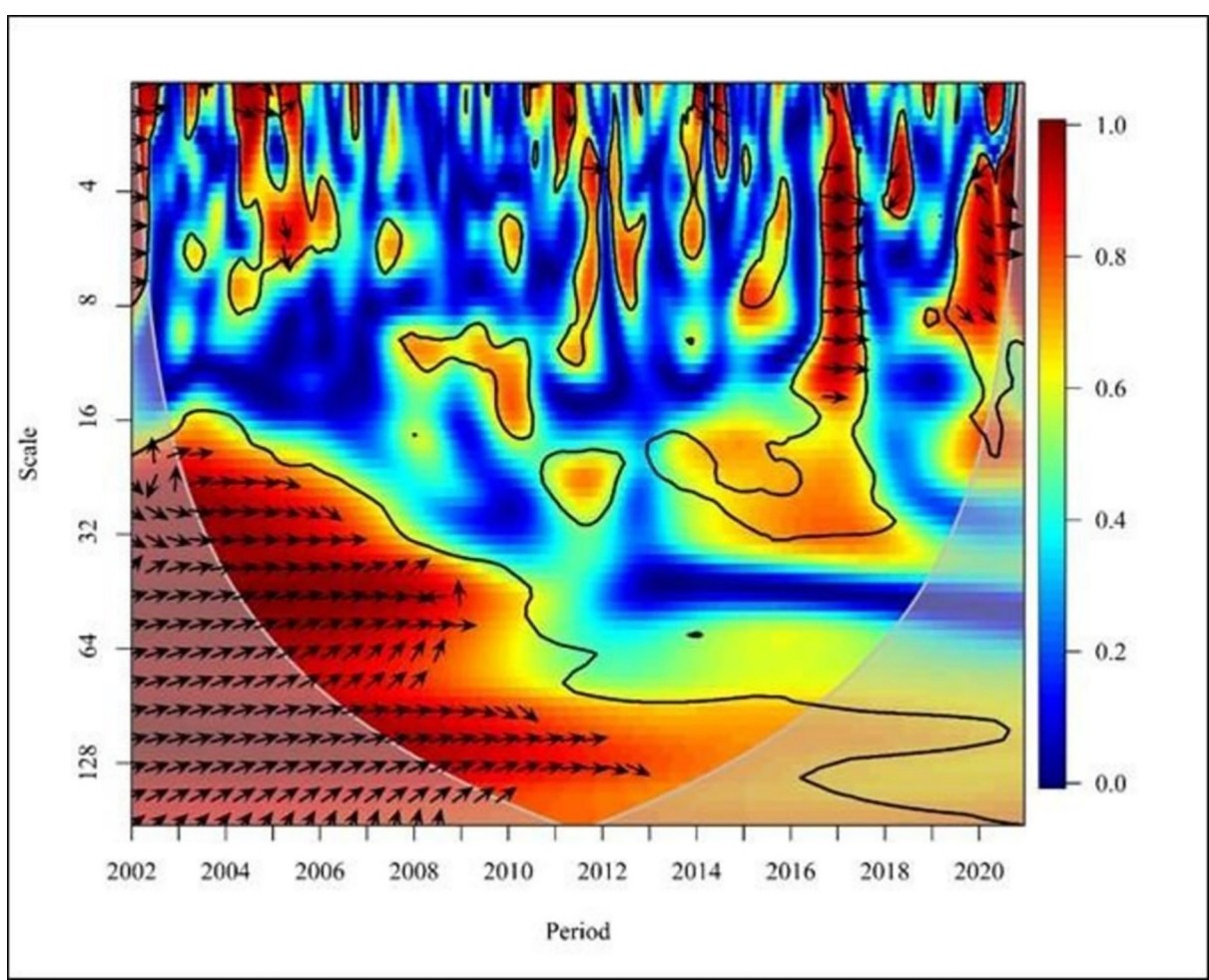

**Fig 10. Wavelet coherence: NE vs FDI for Asia & Oceania continents.** Source: Authors' illustrations using R-Software.

FDI using the VECM model. Additionally, in 2020, there is a bidirectional relationship between FDI and NE with high frequency. Furthermore, the arrows point up and down, depicting a bidirectional rightward trend between FDI and NE with low and medium frequencies from 2004 to 2013. The results of this study support the results of previous studies [39,47], which showed a bidirectional causality between FDI and NE. It confirms that the FDI affects the increase in the export of products, and the flow of FDI is further promoted after the export is increased. Additionally, it is confirmed that there is a positive bidirectional relationship between FDI and exports in the Chinese context, consistent with the above findings [5,38].

## Robustness check

**Panel granger causality test.**   This section presents an overview of the results obtained using the panel Granger causality tests for each region. The results of the Granger causality test for Europe, Africa, North America, South America, Asia & Oceania are presented in Table 4, which was created using the statistical software STATA to analyse the data. Before conducting the Granger causality analysis, a unit root test was carried out to determine whether the variables related to NE and FDI were stationary. The findings of the unit root test, conducted

**Table 4. Granger causality test: NE vs FDI for global context.**

| Classification | NE → FDI | FDI → NE | Causality Findings |
|---|---|---|---|
| Global | 24.985*** | 18.846*** | FDI ↔ NE |
| Africa | 19.584*** | 9.371* | FDI ↔ NE |
| Asia & Oceania | 7.432* | 10.191** | FDI ↔ NE |
| Europe | 9.260* | 11.716** | FDI ↔ NE |
| North America | 13.037** | 6.715* | FDI ↔ NE |
| South America | 9.260** | 8.411** | FDI ↔ NE |

Note: The symbols

*, ** and *** represents 10%, 5% and 1% significance level, respectively.

Source: Authors' calculation based on data from The World Bank [18].

The panel granger causality analysis was carried out for further enhancement of the results.

using the Levin Lin Chu test in 2002, are shown in S3 Appendix. Additionally, the results of the optimal lag length selection criteria are displayed in S4 Appendix.

Table 4 presents a summary of the panel Granger causality results. The findings corroborate the results obtained from Wavelet analysis for regional analysis. In the global context, the results indicate a bidirectional causality between NE and FDI. Furthermore, Table 5 provides a summary of the findings from both the Wavelet coherence and Granger causality analysis for NE and FDI.

The Wavelet coherence technique, which captures the time dependence of variables, aligns with the collective capture of the Granger causality approach. The findings from both models are consistent, providing strong justifications for the study. Both the Granger causality and Wavelet coherence techniques indicate a bidirectional linkage between NE and FDI in each region individually as well as globally. This demonstrates that FDI and NE are interconnected, and the relationship operates from both directions.

**The Granger causality test for cross-country analysis.** This section provides an overview of the results obtained using the Granger causality tests for each country within each continent. The results of the Granger causality test results for Europe, Africa, North America, South America, and Asia & Oceania are presented in Tables 6–10. In addition, the unit root test was conducted to determine whether the variables related to NE and FDI were stationary before performing Granger causality analysis. The findings of the unit root test, reportedly carried out by Dickey-Fuller Phillips and Pierre Perron are shown in S3 Appendix. The results of the optimal lag length selection criteria are also displayed in S4 Appendix.

For the European countries (Table 6), various patterns of causality between FDI and NE were observed. Specifically, Belarus, Bulgaria, Finland, Iceland, North Macedonia, Slovak

**Table 5. Summary findings of Wavelet coherence and Granger causality between NE and FDI.**

| Classification | Wavelet Coherence | Granger Causality |
|---|---|---|
| Global | Bi-directional | Bi-directional |
| Africa | Bi-directional | Bi-directional |
| Asia & Oceania | Bi-directional | Bi-directional |
| Europe | Bi-directional | Bi-directional |
| North America | Bi-directional | Bi-directional |
| South America | Bi-directional | Bi-directional |

Source: Authors' calculation based on data from The World Bank [18].

**Table 6. Test results for granger causality test: NE vs FDI for european region.**

| Country | NE→FDI | FDI→NE | Country | NE→FDI | FDI→NE |
|---|---|---|---|---|---|
| Albania | 15.854*** | 13.708*** | Malta | 0.85684 | 0.2703 |
| Belarus | 3.9145 | 14.667*** | Moldova | 8.7901** | 6.7682* |
| Bulgaria | 2.1121 | 32.46*** | Netherland | 4.87 | 5.6253 |
| Croatia | 12.282*** | 19.293*** | North Macedonia | 0.5741 | 18.628*** |
| Cyprus | 6.4677** | 11.553*** | Norway | 7.4569* | 0.8517 |
| Czech Republic | 1.2465 | 4.8804 | Poland | 2.1832 | 2.6964 |
| Denmark | 7.373* | 17.613*** | Portugal | 36.286*** | 8.8389** |
| Estonia | 1.5542 | 6.2352 | Romania | 25.416*** | 8.8655 |
| Finland | 2.7415 | 9.2809** | Russia | 8.8738 | 3.8568 |
| France | 8.8765** | 7.8321* | Slovak republic | 2.8094 | 6.8662* |
| Germany | 5.465 | 3.8799 | Slovenia | 7.5653* | 7.2115* |
| Greece | 11.607*** | 6.6094 | Spain | 9.6129** | 8.6116** |
| Hungary | 12.224*** | 1.8207 | Sweden | 0.21162 | 9.0669** |
| Iceland | 3.3845 | 8.0771** | Switzerland | 10.716** | 4.0903 |
| Italy | 9.4582** | 2.7612 | Turkey | 0.55143 | 5.4787 |
| Latvia | 10.421** | 22.613*** | UK | 8.058** | 10.602** |
| Lithuania | 6.0971 | 4.3054 | Ukraine | 65.108*** | 35.589*** |

Note: The symbols

*, ** and *** represents 10%, 5% and 1% significance level, respectively.

Source: Authors' calculation based on data from The World Bank [18].

republic, and Sweden showed a unidirectional relationship from FDI to NE. On the other hand, Albania, Croatia, Cyprus, Denmark, France, Latvia, Moldova, Portugal, Slovenia, Spain, the UK, and Ukraine displayed a bidirectional causality between FDI and NE. However, in the remaining European countries, there was no significant directionality between FDI and NE. Overall, 25 out of 34 European countries showed either a unidirectional or bidirectional relationship between FDI and NE, with findings consistent with previous studies in Poland and the Czech Republic [19].

In the African continent (Table 7), the Granger causality test results varied among the 21 countries analysed. Countries like Morocco, Namibia, Seychelles, and Tunisia exhibited a unidirectional causality from FDI to exports, while Cabo Verde, Eswatini, and Zambia showed a unidirectional causality from exports to FDI. Bidirectional causality was observed in five countries (Botswana, Ethiopia, Madagascar, Malawi, and Nigeria). Nine countries (Angola, Djibouti, Egypt, Ghana, Kenya, Mauritius, Sao Tome and Principe, Sierra Leone, and South Africa) showed no significant causality between FDI and NE. The results obtained for Nigeria in this study are similar to previous empirical evidence, indicating a bidirectional connection between FDI and exports. The results suggested that export-promoting measures and reforms in SSA economies led to increased exports and attracted FDI, confirming the relationship observed through the Granger causality test.

When conducting separate Granger causality tests to examine the relationship between FDI and NE in North American countries (Table 8), it was observed that, apart from the Bahamas, Costa Rica, Dominican Republic, Guatemala, Jamaica, Nicaragua, St. Lucia, and the United States, the other eight countries exhibited a modest effect between these variables. Specifically, Canada, Panama, St. Vincent, and the Grenadines showed only unidirectional causality from the FDI to exports. On the contrary, Antigua, and Barbuda, Belize, and Grenada demonstrated

**Table 7. Test results for Granger causality test: NE vs FDI for African region.**

| Country | NE→FDI | FDI→NE | Country | NE→FDI | FDI→NE |
|---|---|---|---|---|---|
| Angola | 1.7137 | 3.1282 | Mauritius | 1.3799 | 4.7617 |
| Botswana | 11.404** | 6.5397* | Morocco | 2.8982 | 8.3688** |
| Cabo Verde (Cape Verde) | 10.437** | 0.26194 | Namibia | 4.8315 | 13.832*** |
| Djibouti | 2.3159 | 3.1687 | Nigeria | 21.4*** | 8.9619** |
| Egypt, Arab Rep. | 1.1146 | 3.4217 | Sao Tome and Principe | 0.75595 | 1.2857 |
| Eswatini (Swaziland) | 9.3748** | 2.3172 | Seychelles | 1.4862 | 14.222*** |
| Ethiopia | 29.086*** | 7.5677* | Sierra Leone | 1.2182 | 1.8236 |
| Ghana | 2.9688 | 0.68038 | South Africa | 4.381 | 3.4413 |
| Kenya | 2.6843 | 6.0063 | Tunisia | 1.9034 | 7.0849* |
| Madagascar | 19.058*** | 10.714** | Zambia | 6.5058* | 3.8498 |
| Malawi | 10.259** | 16.116*** | | | |

Note: The symbols

*, ** and *** represents 10%, 5% and 1% significance level, respectively.

Source: Authors' calculation based on data from The World Bank [18].

unidirectional causality from exports to FDI. Only three countries, namely Dominica, Mexico, and Trinidad and Tobago, displayed a bidirectional relationship between FDI and NE.

The most accurate indicator of manufactured goods' exports is the ratio of total GDP [23]. This criterion plays a crucial role in building an integrated economy while also supporting each country's strong exports. In line with this, the study utilised GMM methods to indicate that the border regions of the Mexican state attract more foreign investment for manufacturing exports, and FDI has the highest impact on exports. However, recent Granger causality results from a different study indicated a different perspective, suggesting a bidirectional relationship between the two variables.

When examining the relationship between FDI and exports in South American countries (Table 9) separately, it was confirmed that Chile alone showed no causality. However, Granger causality was demonstrated in nine out of ten countries in this region. Specifically, Argentina, Bolivia, Guyana, and Peru exhibited a bidirectional causality between FDI and NE. Additionally, Brazil, Colombia, and Ecuador showed unidirectional causality from FDI to exports,

**Table 8. Test results for Granger causality test: NE vs FDI for North American region.**

| Country | NE→FDI | FDI→NE | Country | NE→FDI | FDI→NE |
|---|---|---|---|---|---|
| Antigua and Barbuda | 45.98*** | 5.1583 | Jamaica | 3.5449 | 5.511 |
| Bahamas | 4.4188 | 3.0159 | Mexico | 10.646** | 6.6533* |
| Belize | 33.406*** | 2.1065 | Nicaragua | 4.6039 | 0.93662 |
| Canada | 4.7086 | 29.531*** | Panama | 2.957 | 8.2608** |
| Costa Rica | 1.5058 | 2.6785 | St. Lucia | 3.009 | 3.4747 |
| Dominica | 27.477*** | 22.151*** | St. Vincent and the Grenadines | 5.1691 | 24.867*** |
| Dominican Republic | 1.9998 | 0.73323 | Trinidad and Tobago | 12.871*** | 14.103*** |
| Grenada | 14.934*** | 3.2118 | United States | 0.10454 | 0.9912 |
| Guatemala | 4.2774 | 2.4797 | | | |

Note: The symbols

*, ** A 10%, 5% and 1% significance level, respectively.

Source: Authors' calculation based on data from The World Bank [18].

**Table 9. Test results for Granger causality test: NE vs FDI for South American region.**

| Country | NE→FDI | FDI→NE | Country | NE→FDI | FDI→NE |
|---|---|---|---|---|---|
| Argentina | 8.5887** | 6.9101* | Ecuador | 2.232 | 12.985*** |
| Bolivia | 7.3459* | 7.5015* | Guyana | 58.108*** | 20.507*** |
| Brazil | 4.6122 | 6.6713* | Paraguay | 6.273* | 0.4267 |
| Chile | 2.9319 | 1.6646 | Peru | 6.3315* | 6.6755* |
| Colombia | 1.905 | 8.4016** | Uruguay | 7.1422* | 3.9576 |

Note: The symbols

*, ** and *** represents 10%, 5% and 1% significance level, respectively.

Source: Authors' calculation based on data from The World Bank [18].

while Paraguay and Uruguay showed the opposite direction, indicating unidirectional causality from exports to FDI.

The results obtained for Brazil in this region are consistent with the empirical evidence [22]. Despite delays in attracting FDI in the past three years, exports have shown an increase in both the long and short term. This increase in exports has also contributed to the inflow of FDI, playing a significant role in boosting export-oriented policies in Brazil. The Granger causality test further supports the notion that FDI affects exports in this context.

In the Asia & Oceania regions (Table 10), it was revealed that in 9 out of the 28 countries, there is no Granger causality between FDI and NE. These countries include Australia, Azerbaijan, India, Indonesia, Kazakhstan, Mongolia, Singapore, Thailand, and Vietnam. On the other hand, six countries (Bangladesh, Cambodia, Japan, Saudi Arabia, Solomon Islands, and Vanuatu) showed FDI causality to exports, indicating a unidirectional causality from FDI to exports. In contrast, six countries (Israel, Korea Republic, Kyrgyz Republic, Malaysia, Maldives, and Oman) showed unidirectional causality from exports to FDI in the opposite direction.

Furthermore, seven countries, namely Armenia, China, Kuwait, Nepal, Pakistan, Philippines, and Sri Lanka, demonstrated a bidirectional causal relationship between FDI and NE. The overall result for this region showed a predominantly bidirectional relationship. More than half of the countries in this region appear to have Granger causality. A Chinese study [38]. confirmed a similar finding to the above results, indicating a bidirectional causal relationship between exports and FDI. Moreover, using the Granger causality test, it proved the same result for China in the Asia & Oceania region [5].

The positive two-way relationship between FDI and NE suggests that exports make China's FDI more attractive, and FDI, in turn, contributes to China's export promotion [5]. On the other hand, from a macroeconomic perspective, export expansion indicates a nation's level of global competitiveness [77]. The advantages provided by the government contribute to its competitiveness and thus encourage investment in the country by MNEs. These comparative advantages show that since 1978, China's export growth has increased by 4.5 times the rate of global export growth, which attracts FDI to China. The relationship between FDI and exports also reflects Chinese FDI policies, which have a general preference for encouraging export oriented FDI. In other words, the Chinese government actively promotes export oriented FDI by providing special tax breaks, cheap land use taxes, and other infrastructure services. Furthermore, the effect of FDI on Bangladesh's exports is confirmed using GMM and ARDL methods, which further indicates the previous findings to be consistent with the results obtained in Granger causality [44].

**Table 10. Test results for Granger causality test: NE vs FDI for Asia & Oceania region.**

| Country | NE→FDI | FDI→NE | Country | NE→FDI | FDI→NE |
|---|---|---|---|---|---|
| Armenia | 7.6671* | 10.592** | Malaysia | 142.76*** | 1.0378 |
| Australia | 1.7767 | 2.9081 | Maldives | 13.786*** | 1.0801 |
| Azerbaijan | 0.94975 | 3.0091 | Mongolia | 4.4525 | 5.325 |
| Bangladesh | 2.5857 | 20.069*** | Nepal | 51.524*** | 10.862** |
| Cambodia | 4.189 | 14.978*** | Oman | 10.87** | 1.9509 |
| China | 17.488*** | 9.1691** | Pakistan | 27.367*** | 13.764*** |
| India | 4.9001 | 2.8678 | Philippines | 10.471** | 10.325** |
| Indonesia | 0.8986 | 1.9066 | Saudi Arabia | 5.6184 | 17.597*** |
| Israel | 7.0212* | 8.0949 | Singapore | 4.5997 | 5.0564 |
| Japan | 5.6188 | 11.547*** | Solomon Islands | 3.5406 | 9.7** |
| Kazakhstan | 1.5651 | 2.8407 | Sri Lanka | 27.986*** | 7.6569* |
| Korea Republic | 9.8314** | 1.5139 | Thailand | 2.2313 | 1.798 |
| Kuwait | 9.0515** | 14.086*** | Vanuatu | 0.77888 | 27.734*** |
| Kyrgyz Republic | 10.667** | 1.3457 | Vietnam | 3.1247 | 2.461 |

Note: The symbols

*, ** and *** represents 10%, 5% and 1% significance level, respectively.

Source: Authors' calculation based on data from The World Bank [18].

## Conclusion, limitations and future research

### Conclusion and policy recommendation

The study aimed to understand the relationship between FDI and NE, analysing panel data from 110 countries over 2002 to 2020 using the Wavelet coherence technique and Granger causality test. Unlike previous research limited to specific regions and methodologies, this study provided a global analysis of the causality and correlation between FDI and NE. Empirical findings revealed a bidirectional positive relationship between FDI and NE in the short and medium term globally. The study also classified countries as developed and developing, showing mixed relationships in the short term and no relationship in the long term for developed nations. In the European region, a mixed relationship was evident in the short and medium term, but no relationship was observed in the long term. In Africa, a negative relationship existed, with FDI leading to NE. North America displayed a negative bidirectional causality between FDI and NE, while South America showed a negative relationship in the long run with no causality in the short run from 2002 to 2013. In the Asian and Oceania region, a positive relationship between FDI and NE was evident, with no causality in the short term from 2002 to 2015. Overall, the study demonstrated a mixed and bidirectional relationship between FDI and NE across regions over 19 years, with the Wavelet coherence technique ensuring accuracy.

To further confirm the accuracy of the results obtained from the Wavelet coherence technique, a cross-country analysis was carried out using the Granger causality test, revealing mixed results, including unidirectional, bidirectional, and no causality relationships between FDI and NE. The findings have important implications for global institutions and policymakers, suggesting the need for effective policies that consider the bidirectional influence of FDI and NE. Policymakers can learn from the observed patterns in this study to develop more effective policies to achieve their objectives. While most countries have shown a positive bidirectional causality between exports and FDI, it is important to note that the American and African regions experienced a negative impact of FDI on NE.

## Policy recommendation for different regions

In Africa, diversifying exports beyond primary commodities can attract significant FDI inflows and support economic development. Investments in sectors with high export potential and infrastructure development to enhance trade connectivity are crucial. Policy reforms to improve the investment climate, governance, and regulatory frameworks can also attract FDI. Countries like Ethiopia and Rwanda can serve as successful examples of implementing export-oriented policies to attract FDI and achieve export growth in specific sectors.

Asia & Oceania can benefit from focusing on innovation and technology transfer to boost net exports and attract high-quality FDI. Investment in research and development, fostering collaboration between academia and industry, and regional economic integration initiatives like the ASEAN Economic Community and the CPTPP can facilitate trade flows and attract FDI.

Europe can prioritise export-oriented industries and support SMEs to enhance net exports and attract FDI. Providing access to finance, export promotion schemes, and streamlining regulations for SMEs can boost their export capabilities. Uplifting the investment climate, harmonising investment frameworks, and reducing bureaucratic obstacles can attract precious FDI inflows, as observed in countries like Ireland and Estonia.

In North America, maintaining an open and predictable trade environment is crucial to promote net exports and attract FDI. Trade liberalisation efforts, advocacy for free trade relationships, and the prevention of trade barriers can facilitate NE. Investments in innovation and technology to increase competitiveness and encouraging research and development can attract FDI and drive NE growth.

South America can focus on sustainable development practices to attract responsible FDI and promote NE growth. Implementing environmental regulations, investing in renewable energy projects, and sustainable agriculture practices can attract green investments. Strengthening regional integration through trade agreements like Mercosur can increase market access, facilitate trade flows, and attract FDI, contributing to NE growth.

## Limitations and future research

The study has some limitations, including the use of secondary data that may have measurement errors or data limitations. The fixed time frame of 19 years might not capture long-term changes in the relationship between FDI and NE. The analysis focused on macroeconomic variables and did not consider micro-level factors that could influence FDI and NE. The study also did not account for geopolitical and global economic events that may have impacted the relationship between FDI and NE.

Future research can address these limitations by incorporating more recent data to capture evolving relationships between FDI and NE. Examining micro-level factors and firm-level data can provide deeper insights into the mechanisms driving FDI and NE. Studying the impact of geopolitical events and global economic crises on FDI and NE can offer valuable insights for policymakers. Case studies in specific countries or regions can provide context-specific insights into the relationship between FDI and NE. Additionally, exploring the role of technological advancements and digitalisation in influencing FDI and NE could be a relevant area of future research. By addressing these areas, future research can contribute to a more comprehensive understanding of the complex relationship between FDI and NE and inform more effective policy strategies for sustainable economic growth.

## Supporting information

**S1 Appendix. Summary of the continents and the countries.**
(DOCX)

**S2 Appendix. Data file.**
(XLSX)

**S3 Appendix. Unit root test results.**
(DOCX)

**S4 Appendix. Lag length criteria results.**
(DOCX)

## Author Contributions

**Conceptualization:** Chanaka Fernando, Gayan Vidyapathirana, Ruwan Jayathilaka, Sumudu Munasinghe.

**Data curation:** Sanduni Lakshani, Chanuka Sandaruwan, Chanaka Fernando, Gayan Vidyapathirana.

**Formal analysis:** Sanduni Lakshani, Chanuka Sandaruwan, Ruwan Jayathilaka.

**Investigation:** Sanduni Lakshani.

**Methodology:** Sanduni Lakshani, Ruwan Jayathilaka.

**Software:** Chanuka Sandaruwan, Chanaka Fernando, Gayan Vidyapathirana.

**Supervision:** Ruwan Jayathilaka, Sumudu Munasinghe.

**Validation:** Sanduni Lakshani, Ruwan Jayathilaka.

**Visualization:** Sanduni Lakshani, Chanuka Sandaruwan, Ruwan Jayathilaka.

**Writing – original draft:** Sanduni Lakshani, Chanuka Sandaruwan, Chanaka Fernando, Gayan Vidyapathirana, Ruwan Jayathilaka, Sumudu Munasinghe.

**Writing – review & editing:** Ruwan Jayathilaka, Sumudu Munasinghe.

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
