## [Decision Letter · Decision Letter 0]

4 Jul 2023

PONE-D-23-12176Net-Export and FDI Nexus: Global Study Based on Wavelet Coherence Technique and Granger CausalityPLOS ONE

Dear Dr. Jayathilaka,

Thank you for submitting your manuscript to PLOS ONE. After careful consideration, we feel that it has merit but does not fully meet PLOS ONE’s publication criteria as it currently stands. Therefore, we invite you to submit a revised version of the manuscript that addresses the points raised during the review process.

We look forward to receiving your revised manuscript.

Kind regards,

Fiza Qureshi, PhD

Academic Editor

PLOS ONE

Reviewers' comments:

Reviewer's Responses to Questions

**Comments to the Author**

1. Is the manuscript technically sound, and do the data support the conclusions?

Reviewer #1: No

Reviewer #2: Yes

2. Has the statistical analysis been performed appropriately and rigorously? 

Reviewer #1: Yes

Reviewer #2: Yes

3. Have the authors made all data underlying the findings in their manuscript fully available?

Reviewer #1: No

Reviewer #2: Yes

4. Is the manuscript presented in an intelligible fashion and written in standard English?

Reviewer #1: No

Reviewer #2: Yes

5. Review Comments to the Author

Reviewer #1: The study uses the wavelet method to examine the time-varying interrelation between Net export and FDI. This paper needs to be improved in different ways before it can be accepted. Following are the comments to improve this draft:

1. Recast the selected paper with four parts: the policy hypothesis is first indicated, then the model (theoretical frame) is developed, next the empirical validation confirms or rejects, then produce a set of policy advisory. In other words, a substantial policy discussion is needed, making sure that the hypothesis testing is policy guided and not guided by the objective to verify the validity/applicability of a given statistical tool.

2. It’s not clear whether original data indices are used or any transformation is applied to indices before applying wavelet approach.

3. This article provides some figures about empirical results. However, their information is not clear. Please add a note for the readers and explain how to interpret the wave patterns in the graph.

4. This study applies the wavelet approach to estimate the linkages between various variables. Why Wavelet approach is preferable to other competitive models? Explain it in detail. The method of wavelet significance testing is not discussed in the paper?

5. The biggest problem is just showing the empirical results. The empirical results should be discussed in a better way, and their similarities and dissimilarities with previous studies should be elaborated. More explanation is needed regarding the main findings, especially the difference of results across scales.

6. The Empirical Results should consist of sub-headings, i.e. Preliminary findings, Empirical findings and Robustness check.

7. The manuscript needs to be checked by a native English speaker for style, grammatical errors, typos, spelling and incorrect word choice. There are many problems in format and I suggest the authors should check the whole paper and correct them. The language is not always fluent. There are also many typos and the language is repetitive (throughout the paper).

8. There are some related papers that should be cited. Qureshi, F., Qureshi, S., Vo, X. V., & Junejo, I. (2021). Revisiting the nexus among foreign direct investment, corruption and growth in developing and developed markets. Borsa Istanbul Review, 21(1), 80-91. & Kamal, M. A., Ullah, A., Qureshi, F., Zheng, J., & Ahamd, M. (2023). China’s outward FDI and environmental sustainability in belt and road countries: does the quality of institutions matter?. Journal of Environmental Planning and Management, 66(5), 1002-1036.

9. Policy implications should be improved for the interest of regulators, investors, and policymakers.

10. The literature review section is very comprehensive but the latest literature of 2022 is missing, therefore it is advised to add few recent studies in this section as well.

11. In the literature review section, the author(s) could cite more references on the approaches to describing interrelationships, and particularly the wavelets approach, which is one of the main contributions of this paper.

12. More explanation and justification are needed regarding the choice of the sample period, and the frequency of the data.

13. The introduction section is slow and should be revised in order to provide a more comprehensive background and motivation on the topic and the necessity to consider the interlinkages in the FDI and Net exports. In this regard, the authors should spend more ink to better explain their added value and contribution of the paper to the academic literature, including the added value of using multiscale-based approach.

14. The title should be changed to Net Export and FDI Nexus: Global study based on Wavelet analysis

15. Make sure that all cited papers are presented in the reference list and vice-versa.

Good luck with the revision.

Reviewer #2: The topic is interesting, and the author has done a great job in realizing the subject. However, there are few areas on the paper that is still lagging and should be addressed properly.

Abstract

1. The authors should motivate the choice of variables with theory and empirical backing on the subject

2. Keywords should be revised to match key element of title

3. Rewrite the title to be more catchy

4. Introduction

1. The objective of the paper presented need more clarifications to suit reader to understand the main idea of the paper especially for the study case is needed

2. Literature review

The literature is well written. However, there is need for more recent studies ranging from 2018-2022 to motivate the study properly. The entire study is too scanty and the related literature is not exhausted

Trade openness, FDI, and income inequality: Evidence from sub‐Saharan Africa. African Development Review, 33(1), 193-203

Causal interactions among tourism, foreign direct investment, domestic credits, and economic growth: evidence from selected Mediterranean countries. Portuguese Economic Journal, 19, 195-212

Methodology

1. This section is generally well motivated, Kindly take note of the following minor additions

2. More benefit of the various techniques utilized should be stated

3. The authors should avoid much mathematical expressions or take some to appendix and make the study reader friendly for other practitioner other than academic with out compromise for study intend and quality.

Discussion

1. The discussion is well written, but the authors should like their findings to the previous studies in the literature.

2. There is need for professional proofreading or consult English native support

3. Conclusion

1. The sub-title should be conclusion and policy recommendation but not only conclusion

2. The policy which is the engine of the study is weak and small. I therefore encourage the authors to elaborate more on the policy recommendations to policy makers for the investigated bloc

3. The authors should add limitation of the study and future recommendation

6. PLOS authors have the option to publish the peer review history of their article (what does this mean?). If published, this will include your full peer review and any attached files.

Reviewer #1: No

Reviewer #2: **Yes: **Festus Victor Bekun

---

## [Author Response · Author response to Decision Letter 0]

21 Jul 2023

Point by point response to editor and reviewers.

We would like to express our profound appreciation to the editor and the reviewers for the valuable comments and suggestions made on our manuscript which were very helpful in revising and improving it. Please note that the line numbers referred in this document is aligned with the revised manuscript which has track changes.

Reviewer 1 comment 1: Recast the selected paper with four parts: the policy hypothesis is first indicated, then the model (theoretical frame) is developed, next the empirical validation confirms or rejects, then produce a set of policy advisory. In other words, a substantial policy discussion is needed, making sure that the hypothesis testing is policy guided and not guided by the objective to verify the validity/applicability of a given statistical tool.

Authors’ Response to Reviewer 1 comment 1: Thank you for your comment. We have revised the manuscript based on your suggestion. The revised version includes a more comprehensive discussion of the theoretical framework and an elaboration of the model from line 192 to line 252.

A substantial policy discussion is in the conclusion and policy recommendation section. The revised version can be found starting from line 1288 to line 1377 We appreciate your feedback and the opportunity to improve our work.

“The linkage between NE and FDI is based on theoretical motives that help explain their interactions. These theories provide frameworks for confirming the dynamics between NE and FDI and reveal their underlying mechanisms.

How FDI Affects Net Export

According to the backward linkage theory, FDI creates backward relationships with local suppliers and expands their export capacities. The impact of FDI on backward linkages in Indonesia was examined and found that FDI inflows lead to increased demand for domestic inputs, which in turn support export-oriented industries and stimulate NE [25].

Knowledge and technology transfer theory suggests that FDI helps to transfer knowledge, technology, and managerial competence from investing companies to the host country. This transfer uplifts the productivity and competitiveness of local companies, enabling them to manufacture higher-quality products and services for export [26]. An empirical study revealed the effect of FDI on knowledge transfer and export performance in developing countries. The results indicated that inflows of FDI were positively linked with technological spillovers and improvements in export capabilities, thereby facilitating the idea that knowledge and technology transfer through FDI can enhance NE [26].

Productivity theory argues that FDI prompts productivity growth in the host country, guiding a positive impact on NE. By proposing new technologies, manufacturing techniques, and management practices, FDI uplifts the efficiency and productivity of local industries. A study investigated the linkage between FDI, productivity, and NE in China, revealing that inflows of FDI led to enhanced productivity levels in local firms, which in turn resulted in higher NE [27].

How Net Export Affects on FDI

The export-platform theory demonstrates that NE can captivate FDI as companies seek to use the host country as a platform for exporting to alternative markets. A Mexican study examined the linkage between FDI and exports, which supports the theoretical backing. The evidence explored that FDI inflows guided to enhanced exports, particularly in industries with higher foreign ownership, supporting the notion that net exports can play like a magnet for FDI by serving as an export platform [28].

Product life cycle theory posits that FDI is influenced by the life cycle of products. When a product matures and demand increases in international markets, firms may invest in foreign countries through FDI to set up manufacturing facilities near the target markets. In an empirical study, the linkage between product life cycles, FDI, and NE was analysed. The study revealed that as products transitioned from the introduction stage to the maturity stage, FDI enhanced, leading to a shift in net export patterns as production became more internationally dispersed [29].

Internalisation theory suggests that companies invest in foreign countries through FDI to internalise their business operations. NE can influence the decision to internalise business activities via FDI as companies' goal is to secure their existence in foreign markets and have better control over their export operations. An empirical study investigated the effect of internalisation pros on the choices of the locations of multinational enterprises. The empirical findings demonstrated that internalisation advantages like access to foreign markets played an important role in companies' decisions to attract FDI [30].

According to the theoretical evidence, the theoretical framework (Figure 2) was constructed to investigate the relationship between FDI and NE.

Prior empirical studies support the theoretical framework and all factors considered in this study. The theoretical framework clearly depicts the study aim, which is to identify/understand the relationship between NE and FDI. Thus, the study presents the hypothesis to examine the objective. Furthermore, strong theoretical support is evident in the hypothesis of regarding the relationship between FDI and NE. As a result, the study develops a hypothesis for examining the correlation and/or causality between NE and FDI across all regions and different time frequencies.”

From line 1288 to line 1377

“…………The findings have important implications for global institutions and policymakers, suggesting the need for effective policies that consider the bidirectional influence of FDI and NE. Policymakers can learn from the observed patterns in this study to develop more effective policies to achieve their objectives. While most countries have shown a positive bidirectional causality between exports and FDI, it is important to note that the American and African regions experienced a negative impact of FDI on NE.

Policy Recommendation for Different Regions:

In Africa, diversifying exports beyond primary commodities can attract significant FDI inflows and support economic development. Investments in sectors with high export potential and infrastructure development to enhance trade connectivity are crucial. Policy reforms to improve the investment climate, governance, and regulatory frameworks can also attract FDI. Countries like Ethiopia and Rwanda can serve as successful examples of implementing export-oriented policies to attract FDI and achieve export growth in specific sectors.

Asia & Oceania can benefit from focusing on innovation and technology transfer to boost net exports and attract high-quality FDI. Investment in research and development, fostering collaboration between academia and industry, and regional economic integration initiatives like the ASEAN Economic Community and the CPTPP can facilitate trade flows and attract FDI.

Europe can prioritise export-oriented industries and support SMEs to enhance net exports and attract FDI. Providing access to finance, export promotion schemes, and streamlining regulations for SMEs can boost their export capabilities. Uplifting the investment climate, harmonising investment frameworks, and reducing bureaucratic obstacles can attract precious FDI inflows, as observed in countries like Ireland and Estonia.

In North America, maintaining an open and predictable trade environment is crucial to promote net exports and attract FDI. Trade liberalisation efforts, advocacy for free trade relationships, and the prevention of trade barriers can facilitate NE. Investments in innovation and technology to increase competitiveness and encouraging research and development can attract FDI and drive NE growth.

South America can focus on sustainable development practices to attract responsible FDI and promote NE growth. Implementing environmental regulations, investing in renewable energy projects, and sustainable agriculture practices can attract green investments. Strengthening regional integration through trade agreements like Mercosur can increase market access, facilitate trade flows, and attract FDI, contributing to NE growth.”

Reviewer 1 comment 2: It’s not clear whether original data indices are used, or any transformation is applied to indices before applying wavelet approach.

Authors’ Response to Reviewer 1 comment 2: Thank you for your comment. We have taken your pointers into consideration. The paper demonstrates the data sources and the type of data conversion, starting from line 587 to line 588.

“…………The original data were used for the study, and to facilitate the analysis, all figures were converted into USD millions…………”

Reviewer 1 comment 3: This article provides some figures about empirical results. However, their information is not clear. Please add a note for the readers and explain how to interpret the wave patterns in the graph.

Authors’ Response to Reviewer 1 comment 3: Very much appreciate your comment. We have further explained the wavelet interpretation in the revised manuscript from line 794 to 813. Additionally, table 3 (from line 788 to 789) shows the summary of interpretation of the wavelet correlation displaying the correlation/causation between NE and FDI. 

“The plot uses a horizontal axis to represent the time, with the leftmost area indicating the beginning of the stipulated period (2002) and the rightmost area indicating the end of the interval (2020). The vertical axis represents the period with lower frequency bands (higher frequencies) indicated at the top area of the plot, and higher bands (lower frequencies) represented at the bottom. The scale is divided into portions (upper, middle, and lower), indicating correlation for the short, medium, and long term, respectively. The blue (cold) region indicates no correlation, while the warm (red) region denotes correlated variables. The thick black line represents statistically significant areas of coherence with a 95% confidence level.

The arrows in the plot serve two purposes. Firstly, they indicate the correlation between the two-time variables at a specific point. The leftward arrows (←) denote an anti-phase relationship, indicating a negative correlation between the two variables. Conversely the rightward arrows (→) signify an in-phase relationship, indicating a positive correlation. Secondly, the direction of the arrows (upward and downward) shows which time series leads the relationship at that specific point. The downward arrows (↙, ↘) denote that the first time series (NE) leads the second time series (FDI), while the upward arrows (↖, ↗) demonstrate that the second time series (FDI) causes the first time series (NE). Figs 5, 6, 7, 8, 19 and 10 illustrate the causality and correlation between FDI and NE for the period 2002-2020.”

Reviewer 1 comment 4: This study applies the wavelet approach to estimate the linkages between various variables. Why Wavelet approach is preferable to other competitive models? Explain it in detail. The method of wavelet significance testing is not discussed in the paper?

Authors’ Response to Reviewer 1 comment 4: Thank you for your comment. Your suggestions have been taken into account and the revised manuscript now includes an improved explanation of the significance of the Wavelet coherence technique. This is discussed in more detail from line 620 to 653.

“Empirical studies have highlighted significant advantages of the Wavelet coherence technique over other methodologies. For example, in the analysis of the linkage between stock market returns and macroeconomic variables, Wavelet coherence analysis outperformed traditional techniques by providing a better understanding of frequency-dependent relationships [65]. Another study investigating the connection between land surface temperature and landscape patterns demonstrated the superiority of Wavelet coherence analysis in capturing time-varying and scale-specific coherence [66]. These findings underscore the significance of Wavelet cohere analysis in revealing complex and dynamic relationships. Additionally, the techniques was applied to examine inter-annual and inter-decadal oscillations in monthly precipitation extremes and their teleconnections to climate indices, revealing significant coherence at specific frequency bands that conventional techniques failed to capture [67].

Real-world data often exhibits subtle fluctuations that can be crucial for gaining insights. While Fourier analysis can express certain trends, it fails to capture sudden changes effectively. A Wavelet is a rapidly decaying wave with zero mean and limited duration, available in various shapes and sizes [68]. There are two main type of Wavelet transforms: Continuous and discrete which differ based on the scaling and drawing of these waves.

The significance of Wavelet coherence analysis lies in several key aspects. Unlike other linear and nonlinear analysis methods, it does not require data pre-treatment, enabling the rapid filling of data for multiple time intervals. This aids in generating visual representations of the output instead of mere numerical values. Moreover, the analysis considers the data along short, medium, and long-term dimensions, visually presenting complex and extensive data in a simplified manner. This capability allows for a better understanding of the overall patterns and relationships in the data.”

Reviewer 1 comment 5: The biggest problem is just showing the empirical results. The empirical results should be discussed in a better way, and their similarities and dissimilarities with previous studies should be elaborated. More explanation is needed regarding the main findings, especially the difference of results across scales.

Authors’ Response to Reviewer 1 comment 5: Thank you very much for your comment. We appreciate your suggestion, and we have further clarified the main results in the revised manuscript.

Moreover, similarities and dissimilarities with previous studies are included in lines as follows,

From line 851 to 867.

“…………These findings are in line with the results of the study based on 54 developed and developing countries from 1996 to 2018 [51]. They emphasised that greater trade openness (exports and imports of goods and services) in developed countries lead to less trade barriers, and positive FDI inflow, sending positive signals to international investors. The study also confirmed positive relationship between trade openness and FDI inflow…………” 

From line 890 to 906.

“…………Consequently, India's ranking among other nations has risen with the inflow of FDI [3], and empirical evidence supports the positive effect on FDI on India’s export performance. These findings are consistent with the results of other studies [51] …………”

From line 918 to 920.

“…………However, FDI has a positive association with Turkish trade (imports and exports) from 1974 to 2017 [32]. Although this study finds a positive link during certain times, the above finding is negative, therefore the outcome is different. …………” 

From line 933 to 938.

“…………The positive impact of foreign direct investment on Turkey's import of goods and services can be attributed to the lack of raw materials, management, technology, and finance. Therefore, products are created using imported goods and services, and these finished products can then be exported to the world market [32]. Thus, this empirical study has confirmed the claims.”

From line 964 to 967.

“…………The FDI and NE have a negative relationship in Africa as a whole. It indicates that when FDI increases, NE decreases, and when FDI decreases, NE increases. However, most past literature evaluations have found a positive correlation between FDI and NE [8, 33, 35, 36].”

From line 1002 to 1011.

“…………These findings are consistent with a study conducted in Brazil [22]. They revealed that there is an increase in exports in the short term as well as in the long term, while imports increase only in the short term. FDI also increases due to an increase in NE and a positive relationship between FDI and the trade balance has been shown. Exports are rising in Brazil's trade sectors, especially for market-related businesses. Hence, as exports rise, FDI inflows expand. For this reason, market growth will also increase, and thus economic expansion will also increase [20].Thus, according to the above study, there is a positive relationship between FDI and NE in Brazil, consistent with the above findings…………”

From line 1027 to 1037.

“…………The results above demonstrate a positive linkage between FDI and NE between 2003 and 2012. These findings are consistent with the conclusions of the Wavelet method, providing evidence for a positive relationship between FDI and trade (exports and imports) in Sri Lanka [45]. Additionally, these findings are in line with the results of the study conducted in Jordan [50], which pointed out a positive relationship between FDI and foreign trade, with a 1% increase in FDI leading to a 13% increase in exports. This shows that an increase in FDI in Jordan is associated with an increase in exports, indicating that NE increases as FDI increases and NE decreases as FDI decreases…………”

From line 1038 to 1047.

“…………The impact of FDI on India's export performance during the period 1980 to 2017 has shown a negative relationship using ARDL [46]. On the contrary, the same study shows a unidirectional relationship between FDI and NE using Granger causality. Accordingly, this study shows both similarities and dissimilarities, aligning with the result mentioned above. In 2019, the short-term downward indicators to the left suggest that NE leads to FDI. These findings are consistent with a study conducted in Bangladesh [48], which demonstrated a positive unidirectional impact of NE on FDI using the VECM model. Additionally, in 2020, there is a bidirectional relationship between FDI and NE with high frequency…………”

Reviewer 1 comment 6: The Empirical Results should consist of sub-headings, i.e. Preliminary findings, Empirical findings and Robustness check.

Authors’ Response to Reviewer 1 comment 6: Thank you for your comment. We have taken your point into consideration, and as a result, we have divided the Preliminary findings (line 694 to781), Empirical findings (line 783 to 1054), and Robustness check (line 1057 to 1264) into sub-headings from line 694 to line1264.

Reviewer 1 comment 7: The manuscript needs to be checked by a native English speaker for style, grammatical errors, typos, spelling and incorrect word choice. There are many problems in format, and I suggest the authors should check the whole paper and correct them. The language is not always fluent. There are also many typos and the language is repetitive (throughout the paper).

Authors’ Response to Reviewer 1 comment 7: Thank you very much. We have carefully revised the document, and an expert copyeditor has performed a thorough copy edit. We are confident that the corrected document is free from any language, grammar, and punctuation issues. We highly appreciate your valuable contribution to this article.

Reviewer 1 comment 8: There are some related papers that should be cited. Qureshi, F., Qureshi, S., Vo, X. V., & Junejo, I. (2021). Revisiting the nexus among foreign direct investment, corruption and growth in developing and developed markets. Borsa Istanbul Review, 21(1), 80-91. & Kamal, M. A., Ullah, A., Qureshi, F., Zheng, J., & Ahamd, M. (2023). China’s outward FDI and environmental sustainability in belt and road countries: does the quality of institutions matter? Journal of Environmental Planning and Management, 66(5), 1002-1036.

Authors’ Response to Reviewer 1 comment 8: Thank you for comment, and it is well noted. Relevant articles from mentioned studies have been included in the revised manuscript. This is discussed in more detail from line 532 to 543. 

“…………In a global context, a dynamic relationship between FDI, corruption and economic growth in 54 developed and developing countries between 1996 and 2018 was investigated using a Panel vector autoregression (PVAR) model [51]. Trade openness, total credit to the private sector (CPS), and exchange rate volatility (EV) are among the other control variables utilised in the analysis. The findings reveal that corruption control has a negative (positive) effect on foreign investment and economic development in developing (developed) countries, implying that weaker (stronger) institutional quality and more (lower) corruption increase investment and economic progress. According to the study, economic growth and corruption have a positive bidirectional relationship in developing countries but a negative unidirectional relationship in developed countries. In addition, the authors further find that trade openness (exports and imports of goods and services) is positively associated with FDI inflows.”

Reviewer 1 comment 9: Policy implications should be improved for the interest of regulators, investors, and policymakers.

Authors’ Response to Reviewer 1 comment 9: Thank you for your comment. We have carefully considered your suggestions and have added the suggested improvements to the conclusion section of the study. The revised conclusion section from line 1288 to 1377 now provides a more comprehensive summary of policy implications for regulators, investors, and policymakers.

“…………The findings have important implications for global institutions and policymakers, suggesting the need for effective policies that consider the bidirectional influence of FDI and NE. Policymakers can learn from the observed patterns in this study to develop more effective policies to achieve their objectives. While most countries have shown a positive bidirectional causality between exports and FDI, it is important to note that the American and African regions experienced a negative impact of FDI on NE.

Policy Recommendation for Different Regions:

In Africa, diversifying exports beyond primary commodities can attract significant FDI inflows and support economic development. Investments in sectors with high export potential and infrastructure development to enhance trade connectivity are crucial. Policy reforms to improve the investment climate, governance, and regulatory frameworks can also attract FDI. Countries like Ethiopia and Rwanda can serve as successful examples of implementing export-oriented policies to attract FDI and achieve export growth in specific sectors.

Asia & Oceania can benefit from focusing on innovation and technology transfer to boost net exports and attract high-quality FDI. Investment in research and development, fostering collaboration between academia and industry, and regional economic integration initiatives like the ASEAN Economic Community and the CPTPP can facilitate trade flows and attract FDI.

Europe can prioritise export-oriented industries and support SMEs to enhance net exports and attract FDI. Providing access to finance, export promotion schemes, and streamlining regulations for SMEs can boost their export capabilities. Uplifting the investment climate, harmonising investment frameworks, and reducing bureaucratic obstacles can attract precious FDI inflows, as observed in countries like Ireland and Estonia.

In North America, maintaining an open and predictable trade environment is crucial to promote net exports and attract FDI. Trade liberalisation efforts, advocacy for free trade relationships, and the prevention of trade barriers can facilitate NE. Investments in innovation and technology to increase competitiveness and encouraging research and development can attract FDI and drive NE growth.

South America can focus on sustainable development practices to attract responsible FDI and promote NE growth. Implementing environmental regulations, investing in renewable energy projects, and sustainable agriculture practices can attract green investments. Strengthening regional integration through trade agreements like Mercosur can increase market access, facilitate trade flows, and attract FDI, contributing to NE growth.”

Reviewer 1 comment 10: The literature review section is very comprehensive, but the latest literature of 2022 is missing, therefore it is advised to add few recent studies in this section as well.

Authors’ Response to Reviewer 1 comment 10: Thank you for your comments. We have thoroughly revised the manuscript, incorporating the latest literature from 2022 as much as possible from line 508 to 529. The revised version now includes updated references and incorporates relevant studies from the field. We greatly appreciate your suggestion, as it has strengthened the overall scholarly quality of our work.

“A recent study aimed to investigate the relationship between Bangladesh FDI inflows and export performance, taking into account structural breaks, using annual time series data from 1972 to 2019 [48]. Using the VECM model, the authors found a positive unidirectional effect of real exports (REX) on real FDI (RFDI) in Bangladesh. The study further emphasised the use of structural break methods to investigate this matter. Another Bangladeshi empirical evidence reveals the relationship between FDI inflows and export performance in the long run as well as in the short run from 1995 to 2020. Johansen cointegration and VECM methods were used to obtain the results, and the findings confirm that there is a significant statistical relationship between FDI inflows and export performance in the long run [49]. Similarly, in Jordan, scholars investigated how foreign trade (FT), inflation rate (INFR), gross domestic product (GDP), interest rate (IR), and FDI interact with macroeconomic variables using commonly used approaches such as unit root, cointegration, and ARDL [50]. The final data revealed a statistically significant positive linkage between FDI and international trade in Jordan. The authors also discovered that GDP and international FT have a statistically significant positive link, whereas INFR and exports have a statistically significant negative relationship.”

Reviewer 1 comment 11: In the literature review section, the author(s) could cite more references on the approaches to describing interrelationships, and particularly the wavelets approach, which is one of the main contributions of this paper.

Authors’ Response to Reviewer 1 comment 11: Thank you for your valuable feedback. We appreciate your suggestion to cite more references on the approaches to describing interrelationships, specifically the wavelet approach, which is a significant contribution to our paper. We have thoroughly reviewed the literature and have included additional references that highlight the various approaches and the prominence of wavelets in analyzing interrelationships from line 336 to 347. The revised manuscript now provides a more comprehensive discussion on the topic, showcasing the relevance and importance of the wavelet approach. We sincerely thank you for your insightful comment, which has greatly improved the overall quality of our literature review section.

“In another study, annual data from 1970 to 2010, was used to explore the impact of key macroeconomic variables (economic growth, exports, gross capital formation, trade openness, inflation) on FDI in Nigeria [35]. The ARDL technique was used for this study and the Wavelet coherence technique may be used to further confirm the results. Additionally, Fully modified ordinary least square (FMOLS) and Dynamic ordinary least square (DOLS) were utilised to test the robustness of ARDL long-run estimation. According to financial statements from the ARDL long-run estimate, exports and trade openness have a positive effect on FDI data. This study examined data collected over several years longitudinally to gain a better understanding of the relationship between FDI and macroeconomic variables. As a result, a clear relationship between the factors can be revealed. In addition, to the best of the authors' knowledge, no previous research has used Wavelet coherence and Wavelet correlation approaches to investigate these dynamics.”

Reviewer 1 comment 12: More explanation and justification are needed regarding the choice of the sample period, and the frequency of the data.

Authors’ Response to Reviewer 1 comment 12: Thank you for your comment. We have taken your pointers into consideration and revised the manuscript to include a more explanation detailed regarding the choice of the sample period, and the frequency of the data. The revised version can be found from line 567 to line 582.

“…………The data were collected from the World Bank, considering this specific timeframe due to various significant factors that could impact the relationship and variation in FDI and NE. Notably, the global financial crisis and the global oil crisis of 2007-2008 were key factors influencing FDI and NE fluctuations during 2002 to 2022 [52]. Additionally, the years 2019 and 2020 were critical due to significant events impacting the global economy, such as Russia's invasion of Ukraine and the spread of the COVID-19 pandemic [52]. Thus, the period from 2002 to 2022 was chosen to conduct a comparative investigation of the effects of these events on the correlation between NE and FDI.”

Reviewer 1 comment 13: The introduction section is slow and should be revised in order to provide a more comprehensive background and motivation on the topic and the necessity to consider the interlinkages in the FDI and Net exports. In this regard, the authors should spend more ink to better explain their added value and contribution of the paper to the academic literature, including the added value of using multiscale-based approach.

Authors’ Response to Reviewer 1 comment 13: Thank you very much for your comment. We appreciate your suggestion and have revised the manuscript accordingly. The revised version includes a more detailed background and motivation on the topic and the need to consider the interrelationships of FDI and net exports, as well as the importance of these two variables in the study. The revised version can be found starting from line 49 to line 188.

“International trade and foreign direct investment (FDI) are often linked and contribute to a country's economic growth in multiple ways [1, 2]. Moreover, the dynamic relationship between FDI and trade in developing and developed countries is of significant theoretical and empirical interest among scholars [1, 3-9] . Furthermore, the role of both exports and FDI in economic growth has been highlighted in recent literature. 

Global FDI has become a significant phenomenon in recent years due to globalisation, the rapid expansion of international trade, economic interdependence, and inter-regional trade agreements [10]. Moreover, international trade is essential because no economy in the world is entirely self-sufficient. While the export led growth hypothesis postulates that exports are the primarily driver of overall economic development [11], the empirical evidence shows that FDI flows have been expanding at a rate that outpaces the growth in the volume of international trade [12]. However, there is still disagreement and different opinions among the export-led growth literature and the FDI growth literature.

Several theoretical and empirical studies confirm that there is a complementary relationship between FDI and trade, highlighting how FDI inflows positively influence a country's net exports vice-versa [1, 3-8]. However, some scholars argued that FDI and trade react as substitutes in the presence of international trade barriers [9, 13]. This means that there is a negative relationship between FDI and trade. In addition, there are also some empirical studies that have failed to find any significant relationship between FDI and trade [14]. This phenomenon occurs when the returns from foreign investment, such as facilitating access to international markets, enable the transfer of expertise and new technology, increase local capital, and develop local human capital through training and the promotion of export industries in the host country [15]. Hence, the policy decisions relevant to international trade and foreign investment are crucial in determining the future direction of a country.

As the world over the years, FDI inflows have steadily and substantially increased from US$ 751.69 billion in 2002 to US$ 1.28 trillion in 2020 [16]. The impact of FDI inflows on a host country has become a subject of ongoing discussions in recent years. Meanwhile, NE have also substantially increased and reached US$ 22.48 trillion in 2020 [17]. However, when considered region-wise, a striking feature of the last two decades is that the regional behaviour of FDI and NE differs and is not consistent with global trends (Figure 1).

Although many scholars have already widely discussed this comprehensive relationship based on different geographical regions, previous studies were most likely based on a single or a few geographical areas, and still, there is no study found as continent-wise analysis in past literature. Moreover, there are still different opinions and disagreements in past literature, and different results have been shown in different borders and regions. 

A study based on African regions discovers a close link between FDI and NE period for 1980 to 2007, confirming the positive one-way effect of FDI on the balance of trade through export promotion [8]. However, another study offered a completely different perspective on the relationship between FDI and NE based on 8 developing countries in Asia, and the study found that there is a bidirectional linkage between exports and economic growth in the short run, and exports have a long-run effect on FDI from 1986 to 2013 [2]. In another recent empirical study that conducted in European Union countries, it was revealed that FDI investments increase trade flows in the countries being considered, with no signs of a relationship between the variables [19]. 

Some empirical studies in the American region have revealed that the impact of FDI inflows on Latin America's trade balance is positive with a one-way relationship from 1970 to 1994 [20-22]. These findings were validated by another study conducted in Mexican states, revealing that high FDI impacts the export of products in border areas in Mexican states from 2007 to 2015 [23]. Accordingly, it appears that the relationship between FDI and trade cannot be generalised for every region, and the nature of the relationship may change depending on regional cooperation, growth, rules and regulations, policies, and other factors.

On the other hand, the study based on the Organization for Economic Cooperation and Development (OECD) states that the temperament and behaviour of the relationship also change over time, reflecting the complexity of the FDI and trade nexus [24]. The study further revealed that international trade helped to promote more FDI until the middle of the 1980s. However, the direction of the relationship between FDI and international trade changed after this period [24].

According to the past literature, there has been no study covering all regions to compare the results of each region and specific country, and there is a need for a study at the regional level. Therefore, this study provides a detailed explanation of the causality and correlation of FDI on NE across a broad scope at different times, in contrast to prior studies based on a single country or small geographic area.

Accordingly, the study’s main objective is exploring the linkage between NE and FDI inflows, considering variation in different periods over 19 years. This study was undertaken as a continent-wide study to investigate variations across multiple regions concurrently. Furthermore, to enhance comprehension of the relationship between NE and FDI at the country level, a Granger causality test was conducted for each individual country using the annual time series data period for 2002-2020. Accordingly, the study contributes to the advancement of the body of literature in three ways. Firstly, the novel utilisation of the Wavelet coherence approach adds a new perspective to exploring the nature of the relationship and causality between NE and FDI. This method can capture short and medium-term changes in the direction of the relationship during the specified study period, which previous studies could not discern. In addition, even slight changes at long-term, short-term, and medium-term time scales are reflected in these results, which should have been addressed in previous studies. Secondly, the study covers a global perspective, considering over 110 countries from all habitable continents and spanning nearly two decades. Existing literature has focused on specific regions or individual countries, and no comprehensive global study compares all habitable continents. Finally, by conducting a country- wise analysis to examine the behaviour of individual countries to gain further insight into the relationship between NE and FDI, this study contributes a comparative approach to the existing literature.

Moreover, a Granger causality analysis was also carried out to understand further the relationship between NE and FDI in each country. The test results will aid in accurately identifying the effect and its direction between the variables. Hence, a significant justification is provided along with these results by utilising two analysis methods.

The article begins with a review of the findings of previous scholars who have conducted studies on a similar focus while an explanation of the data and methodology used is discussed next. The latter part. Will present the results and discussion, with an overall conclusion of the study.”

Reviewer 1 comment 14: The title should be changed to Net Export and FDI Nexus: Global study based on Wavelet analysis

Authors’ Response to Reviewer 1 comment 14: We appreciate your feedback. In response, we have revised the title to be more engaging and attention-grabbing, while still maintaining accuracy in reflecting the essence of our research We firmly believe that the new title will better capture the interest of readers and potential audiences.

From line 3 to 4.

 “From Short to Long Term: Dynamic Analysis of FDI and Net Export in Global Regions”

Reviewer 1 comment 15: Make sure that all cited papers are presented in the reference list and vice-versa.

Authors’ Response to Reviewer 1 comment 15: We very much appreciate your comment. All papers cited are listed in the reference list and vice-versa.

Reviewer 2 comment 1: The authors should motivate the choice of variables with theory and empirical backing on the subject.

Keywords should be revised to match key element of title.

Rewrite the title to be more catchy

Authors’ Response to Reviewer 2 comment 1: Thank you for your comment. We greatly appreciate your feedback. In response, we have added a more detailed explanation of the choice of variables, backed by both theory and empirical evidence, in the abstract (lines 28 to 35). Additionally, we have further elaborated on the theoretical framework section (line 192 to 252) to provide a clear rationale for the selection of variables.

From lines 28 to 35

“It is crucial to examine the impact between foreign direct investment (FDI) and net exports (NE) for unveiling international trade dynamics, and the economic development of different geographical regions. It yields sharp insights into how FDI inflows, driven by theories such as backward linkage, export platform, and knowledge transfer, enhance a host country's export capacity and contribute to economic growth. Moreover, studying the reciprocal linkages between FDI and NE helps recognise the aspects of domestic factors, such as productivity and the product life cycle, in attracting FDI and increasing export performance.”

2. Thank you for your comment. We appreciate your suggestion. In response, we have revised the keywords to better align with the key elements of the title. The updated keywords now accurately represent the main focus and content of our study.

From line 46 to 47.

“Keywords: Net Export, FDI inflow, Wavelet coherence, Granger causality, Global Regions”

3. We appreciate your suggestion to make the title more catchy. we have revised the title to be more engaging and attention-grabbing, while still accurately reflecting the essence of our research. We believe the new title will better capture the interest of readers and potential audiences.

From line 3 to 4.

 “From Short to Long Term: Dynamic Analysis of FDI and Net Export in Global Regions.”

Reviewer 2 comment 2: The objective of the paper presented need more clarifications to suit reader to understand the main idea of the paper especially for the study case is needed

Authors’ Response to Reviewer 2 comment 2: We very much appreciate your comment and have taken it into account. We agree with your comment. The revised manuscript includes well-explained objectives with more clarifications to suit reader to understand the main idea of the paper from line 49 to 188. 

“International trade and foreign direct investment (FDI) are often linked and contribute to a country's economic growth in multiple ways [1, 2]. Moreover, the dynamic relationship between FDI and trade in developing and developed countries is of significant theoretical and empirical interest among scholars [1, 3-9] . Furthermore, the role of both exports and FDI in economic growth has been highlighted in recent literature. 

Global FDI has become a significant phenomenon in recent years due to globalisation, the rapid expansion of international trade, economic interdependence, and inter-regional trade agreements [10]. Moreover, international trade is essential because no economy in the world is entirely self-sufficient. While the export led growth hypothesis postulates that exports are the primarily driver of overall economic development [11], the empirical evidence shows that FDI flows have been expanding at a rate that outpaces the growth in the volume of international trade [12]. However, there is still disagreement and different opinions among the export-led growth literature and the FDI growth literature.

Several theoretical and empirical studies confirm that there is a complementary relationship between FDI and trade, highlighting how FDI inflows positively influence a country's net exports vice-versa [1, 3-8]. However, some scholars argued that FDI and trade react as substitutes in the presence of international trade barriers [9, 13]. This means that there is a negative relationship between FDI and trade. In addition, there are also some empirical studies that have failed to find any significant relationship between FDI and trade [14]. This phenomenon occurs when the returns from foreign investment, such as facilitating access to international markets, enable the transfer of expertise and new technology, increase local capital, and develop local human capital through training and the promotion of export industries in the host country [15]. Hence, the policy decisions relevant to international trade and foreign investment are crucial in determining the future direction of a country.

As the world over the years, FDI inflows have steadily and substantially increased from US$ 751.69 billion in 2002 to US$ 1.28 trillion in 2020 [16]. The impact of FDI inflows on a host country has become a subject of ongoing discussions in recent years. Meanwhile, NE have also substantially increased and reached US$ 22.48 trillion in 2020 [17]. However, when considered region-wise, a striking feature of the last two decades is that the regional behaviour of FDI and NE differs and is not consistent with global trends (Figure 1).

Although many scholars have already widely discussed this comprehensive relationship based on different geographical regions, previous studies were most likely based on a single or a few geographical areas, and still, there is no study found as continent-wise analysis in past literature. Moreover, there are still different opinions and disagreements in past literature, and different results have been shown in different borders and regions. 

A study based on African regions discovers a close link between FDI and NE period for 1980 to 2007, confirming the positive one-way effect of FDI on the balance of trade through export promotion [8]. However, another study offered a completely different perspective on the relationship between FDI and NE based on 8 developing countries in Asia, and the study found that there is a bidirectional linkage between exports and economic growth in the short run, and exports have a long-run effect on FDI from 1986 to 2013 [2]. In another recent empirical study that conducted in European Union countries, it was revealed that FDI investments increase trade flows in the countries being considered, with no signs of a relationship between the variables [19]. 

Some empirical studies in the American region have revealed that the impact of FDI inflows on Latin America's trade balance is positive with a one-way relationship from 1970 to 1994 [20-22]. These findings were validated by another study conducted in Mexican states, revealing that high FDI impacts the export of products in border areas in Mexican states from 2007 to 2015 [23]. Accordingly, it appears that the relationship between FDI and trade cannot be generalised for every region, and the nature of the relationship may change depending on regional cooperation, growth, rules and regulations, policies, and other factors.

On the other hand, the study based on the Organization for Economic Cooperation and Development (OECD) states that the temperament and behaviour of the relationship also change over time, reflecting the complexity of the FDI and trade nexus [24]. The study further revealed that international trade helped to promote more FDI until the middle of the 1980s. However, the direction of the relationship between FDI and international trade changed after this period [24].

According to the past literature, there has been no study covering all regions to compare the results of each region and specific country, and there is a need for a study at the regional level. Therefore, this study provides a detailed explanation of the causality and correlation of FDI on NE across a broad scope at different times, in contrast to prior studies based on a single country or small geographic area.

Accordingly, the study’s main objective is exploring the linkage between NE and FDI inflows, considering variation in different periods over 19 years. This study was undertaken as a continent-wide study to investigate variations across multiple regions concurrently. Furthermore, to enhance comprehension of the relationship between NE and FDI at the country level, a Granger causality test was conducted for each individual country using the annual time series data period for 2002-2020. Accordingly, the study contributes to the advancement of the body of literature in three ways. Firstly, the novel utilisation of the Wavelet coherence approach adds a new perspective to exploring the nature of the relationship and causality between NE and FDI. This method can capture short and medium-term changes in the direction of the relationship during the specified study period, which previous studies could not discern. In addition, even slight changes at long-term, short-term, and medium-term time scales are reflected in these results, which should have been addressed in previous studies. Secondly, the study covers a global perspective, considering over 110 countries from all habitable continents and spanning nearly two decades. Existing literature has focused on specific regions or individual countries, and no comprehensive global study compares all habitable continents. Finally, by conducting a country- wise analysis to examine the behaviour of individual countries to gain further insight into the relationship between NE and FDI, this study contributes a comparative approach to the existing literature.

Moreover, a Granger causality analysis was also carried out to understand further the relationship between NE and FDI in each country. The test results will aid in accurately identifying the effect and its direction between the variables. Hence, a significant justification is provided along with these results by utilising two analysis methods.

The article begins with a review of the findings of previous scholars who have conducted studies on a similar focus while an explanation of the data and methodology used is discussed next. The latter part. Will present the results and discussion, with an overall conclusion of the study.”

Reviewer 2 comment 3: The literature is well written. However, there is need for more recent studies ranging from 2018-2022 to motivate the study properly. The entire study is too scanty, and the related literature is not exhausted

Trade openness, FDI, and income inequality: Evidence from sub‐Saharan Africa. African Development Review, 33(1), 193-203

Causal interactions among tourism, foreign direct investment, domestic credits, and economic growth: evidence from selected Mediterranean countries. Portuguese Economic Journal, 19, 195-212

Authors’ Response to Reviewer 2 comment 3: Thank you very much for your comments. As suggested, the revised manuscript has been remodeled to include the latest literature from 2018 to 2022, providing a well-motivated study. The relevant additions can be found in the revised manuscript.

From line 336 to 347

“In another study, annual data from 1970 to 2010, was used to explore the impact of key macroeconomic variables (economic growth, exports, gross capital formation, trade openness, inflation) on FDI in Nigeria [35]. The ARDL technique was used for this study and the Wavelet coherence technique may be used to further confirm the results. Additionally, Fully modified ordinary least square (FMOLS) and Dynamic ordinary least square (DOLS) were utilised to test the robustness of ARDL long-run estimation. According to financial statements from the ARDL long-run estimate, exports and trade openness have a positive effect on FDI data. This study examined data collected over several years longitudinally to gain a better understanding of the relationship between FDI and macroeconomic variables. As a result, a clear relationship between the factors can be revealed. In addition, to the best of the authors' knowledge, no previous research has used Wavelet coherence and Wavelet correlation approaches to investigate these dynamics.”

From line 349 to 355.

“Moreover, conducted this study to examine the long-run relationship between FDI and exports using the ARDL model considering the period from 1980 to 2015 [36]. The findings show that FDI in Nigeria has a positive and statistically significant effect on total exports. Further, it appears that FDI in primary sectors and manufacturing sectors has a positive and significant long-run relationship with both total exports and oil exports, but FDI in the services sector has no significant effect on Nigerian exports.”

From line 493 to 505.

“ARDL cointegration is used to analyse the impact of FDI on trade in Sri Lanka for time series data from 1980 to 2016 [45]. The findings of this study, through short-run and long-run estimations demonstrated a substantial positive linkage between the two variables. Additionally, the study emphasises that ARDL and Granger causality test are used to investigate the impact of FDI on India's export performance [46]. The results show that though ARDL confirms a negative effect of FDI on exports in the short run, there is a unidirectional relationship between FDI and exports through Granger causality with the use of data from 1980 to 2017. Using the same methodology, a bidirectional relationship between FDI and exports is also revealed, providing further evidence over the same period [47].”

From line 508 to 529.

“A recent study aimed to investigate the relationship between Bangladesh FDI inflows and export performance, taking into account structural breaks, using annual time series data from 1972 to 2019 [48]. Using the VECM model, the authors found a positive unidirectional effect of real exports (REX) on real FDI (RFDI) in Bangladesh. The study further emphasised the use of structural break methods to investigate this matter. Another Bangladeshi empirical evidence reveals the relationship between FDI inflows and export performance in the long run as well as in the short run from 1995 to 2020. Johansen cointegration and VECM methods were used to obtain the results, and the findings confirm that there is a significant statistical relationship between FDI inflows and export performance in the long run [49]. Similarly, in Jordan, scholars investigated how foreign trade (FT), inflation rate (INFR), gross domestic product (GDP), interest rate (IR), and FDI interact with macroeconomic variables using commonly used approaches such as unit root, cointegration, and ARDL [50]. The final data revealed a statistically significant positive linkage between FDI and international trade in Jordan. The authors also discovered that GDP and international FT have a statistically significant positive link, whereas INFR and exports have a statistically significant negative relationship.”

From line 532 to 543.

“In a global context, a dynamic relationship between FDI, corruption and economic growth in 54 developed and developing countries between 1996 and 2018 was investigated using a Panel vector autoregression (PVAR) model [51]. Trade openness, total credit to the private sector (CPS), and exchange rate volatility (EV) are among the other control variables utilised in the analysis. The findings reveal that corruption control has a negative (positive) effect on foreign investment and economic development in developing (developed) countries, implying that weaker (stronger) institutional quality and more (lower) corruption increase investment and economic progress. According to the study, economic growth and corruption have a positive bidirectional relationship in developing countries but a negative unidirectional relationship in developed countries. In addition, the authors further find that trade openness (exports and imports of goods and services) is positively associated with FDI inflows.”

Reviewer 2 comment 4: This section is generally well motivated, Kindly take note of the following minor additions.

1. More benefit of the various techniques utilized should be stated

2. The authors should avoid much mathematical expressions or take some to appendix and make the study reader friendly for other practitioner other than academic without compromise for study intend and quality.

Authors’ Response to Reviewer 2 comment 4: Thank you for your comment. Your suggestions have been taken into consideration, and the revised manuscript now includes a more comprehensive explanation of the benefits of the utilised technique. This is discussed in more detail from line 620 to 653.

“Empirical studies have highlighted significant advantages of the Wavelet coherence technique over other methodologies. For example, in the analysis of the linkage between stock market returns and macroeconomic variables, Wavelet coherence analysis outperformed traditional techniques by providing a better understanding of frequency-dependent relationships [65]. Another study investigating the connection between land surface temperature and landscape patterns demonstrated the superiority of Wavelet coherence analysis in capturing time-varying and scale-specific coherence [66]. These findings underscore the significance of Wavelet cohere analysis in revealing complex and dynamic relationships. Additionally, the techniques was applied to examine inter-annual and inter-decadal oscillations in monthly precipitation extremes and their teleconnections to climate indices, revealing significant coherence at specific frequency bands that conventional techniques failed to capture [67].

Real-world data often exhibits subtle fluctuations that can be crucial for gaining insights. While Fourier analysis can express certain trends, it fails to capture sudden changes effectively. A Wavelet is a rapidly decaying wave with zero mean and limited duration, available in various shapes and sizes [68]. There are two main type of Wavelet transforms: Continuous and discrete which differ based on the scaling and drawing of these waves.

The significance of Wavelet coherence analysis lies in several key aspects. Unlike other linear and nonlinear analysis methods, it does not require data pre-treatment, enabling the rapid filling of data for multiple time intervals. This aids in generating visual representations of the output instead of mere numerical values. Moreover, the analysis considers the data along short, medium, and long-term dimensions, visually presenting complex and extensive data in a simplified manner. This capability allows for a better understanding of the overall patterns and relationships in the data.”

2. Thank you for your insightful comments and constructive suggestions. We are grateful for the opportunity to improve our manuscript based on your expertise. We have included this idea in these mathematical expressions manuscript to develop the quality and gravity of our methodology. We remain committed to maintaining the highest standards of scholarly research and ensuring the credibility of our findings. we extend our gratitude for your valuable input and the positive impact it has had on our manuscript. We appreciate your commitment to maintaining the integrity and rigor of the research process.

Reviewer 2 comment 5: 1. The discussion is well written, but the authors should like their findings to the previous studies in the literature.

2. There is need for professional proofreading or consult English native support

Authors’ Response to Reviewer 2 comment 5: Thank you for your comment. We have further explained the main results in the revised manuscript.

From line 851 to 867.

“…………These findings are in line with the results of the study based on 54 developed and developing countries from 1996 to 2018 [51]. They emphasised that greater trade openness (exports and imports of goods and services) in developed countries lead to less trade barriers, and positive FDI inflow, sending positive signals to international investors. The study also confirmed positive relationship between trade openness and FDI inflow…………” 

From line 890 to 904.

“…………Consequently, India's ranking among other nations has risen with the inflow of FDI [3], and empirical evidence supports the positive effect on FDI on India’s export performance. These findings are consistent with the results of other studies [51] …………”

From line 918 to 920.

“…………However, FDI has a positive association with Turkish trade (imports and exports) from 1974 to 2017 [32]. Although this study finds a positive link during certain times, the above finding is negative, therefore the outcome is different. …………” 

From line 933 to 938.

“…………The positive impact of foreign direct investment on Turkey's import of goods and services can be attributed to the lack of raw materials, management, technology, and finance. Therefore, products are created using imported goods and services, and these finished products can then be exported to the world market [32]. Thus, this empirical study has confirmed the claims.”

From line 964 to 967.

“…………The FDI and NE have a negative relationship in Africa as a whole. It indicates that when FDI increases, NE decreases, and when FDI decreases, NE increases. However, most past literature evaluations have found a positive correlation between FDI and NE [8, 33, 35, 36].”

From line 1002 to 1011.

“…………These findings are consistent with a study conducted in Brazil [22]. They revealed that there is an increase in exports in the short term as well as in the long term, while imports increase only in the short term. FDI also increases due to an increase in NE and a positive relationship between FDI and the trade balance has been shown. Exports are rising in Brazil's trade sectors, especially for market-related businesses. Hence, as exports rise, FDI inflows expand. For this reason, market growth will also increase, and thus economic expansion will also increase [20].Thus, according to the above study, there is a positive relationship between FDI and NE in Brazil, consistent with the above findings…………”

From line 1027 to 1037.

“…………The results above demonstrate a positive linkage between FDI and NE between 2003 and 2012. These findings are consistent with the conclusions of the Wavelet method, providing evidence for a positive relationship between FDI and trade (exports and imports) in Sri Lanka [45]. Additionally, these findings are in line with the results of the study conducted in Jordan [50], which pointed out a positive relationship between FDI and foreign trade, with a 1% increase in FDI leading to a 13% increase in exports. This shows that an increase in FDI in Jordan is associated with an increase in exports, indicating that NE increases as FDI increases and NE decreases as FDI decreases…………”

From line 1038 to 1047.

“…………The impact of FDI on India's export performance during the period 1980 to 2017 has shown a negative relationship using ARDL [46]. On the contrary, the same study shows a unidirectional relationship between FDI and NE using Granger causality. Accordingly, this study shows both similarities and dissimilarities, aligning with the result mentioned above. In 2019, the short-term downward indicators to the left suggest that NE leads to FDI. These findings are consistent with a study conducted in Bangladesh [48], which demonstrated a positive unidirectional impact of NE on FDI using the VECM model. Additionally, in 2020, there is a bidirectional relationship between FDI and NE with high frequency…………”

Reviewer 2 comment 6: The sub-title should be conclusion and policy recommendation but not only conclusion

2. The policy which is the engine of the study is weak and small. I therefore encourage the authors to elaborate more on the policy recommendations to policy makers for the investigated bloc.

3. The authors should add limitation of the study and future recommendation

Authors’ Response to Reviewer 2 comment 6: 1. Thank you for your comment. We appreciate your suggestion regarding the sub-title. In response, we have revised the sub-title to "Conclusion and Policy Recommendations" instead of just "Conclusion." The revised version can be found starting from line 1268 to line 1377. 

2. We acknowledge your suggestion regarding the policy recommendations and agree that further elaboration is needed. In response, we have expanded and strengthened the policy recommendations section to provide more comprehensive guidance to policymakers. The revised version can be found starting from line 1288 to line 1377. 

“…………The findings have important implications for global institutions and policymakers, suggesting the need for effective policies that consider the bidirectional influence of FDI and NE. Policymakers can learn from the observed patterns in this study to develop more effective policies to achieve their objectives. While most countries have shown a positive bidirectional causality between exports and FDI, it is important to note that the American and African regions experienced a negative impact of FDI on NE.

Policy Recommendation for Different Regions:

In Africa, diversifying exports beyond primary commodities can attract significant FDI inflows and support economic development. Investments in sectors with high export potential and infrastructure development to enhance trade connectivity are crucial. Policy reforms to improve the investment climate, governance, and regulatory frameworks can also attract FDI. Countries like Ethiopia and Rwanda can serve as successful examples of implementing export-oriented policies to attract FDI and achieve export growth in specific sectors.

Asia & Oceania can benefit from focusing on innovation and technology transfer to boost net exports and attract high-quality FDI. Investment in research and development, fostering collaboration between academia and industry, and regional economic integration initiatives like the ASEAN Economic Community and the CPTPP can facilitate trade flows and attract FDI.

Europe can prioritise export-oriented industries and support SMEs to enhance net exports and attract FDI. Providing access to finance, export promotion schemes, and streamlining regulations for SMEs can boost their export capabilities. Uplifting the investment climate, harmonising investment frameworks, and reducing bureaucratic obstacles can attract precious FDI inflows, as observed in countries like Ireland and Estonia.

In North America, maintaining an open and predictable trade environment is crucial to promote net exports and attract FDI. Trade liberalisation efforts, advocacy for free trade relationships, and the prevention of trade barriers can facilitate NE. Investments in innovation and technology to increase competitiveness and encouraging research and development can attract FDI and drive NE growth.

South America can focus on sustainable development practices to attract responsible FDI and promote NE growth. Implementing environmental regulations, investing in renewable energy projects, and sustainable agriculture practices can attract green investments. Strengthening regional integration through trade agreements like Mercosur can increase market access, facilitate trade flows, and attract FDI, contributing to NE growth.”

3. Duly noted on the comment with thanks. The manuscript already includes the limitations and further discussion from line 1386 to 1401, moreover, we have added more limitations and future recommendation, specifically in lines, 

“The study has some limitations, including the use of secondary data that may have measurement errors or data limitations. The fixed time frame of 19 years might not capture long-term changes in the relationship between FDI and NE. The analysis focused on macroeconomic variables and did not consider micro-level factors that could influence FDI and NE. The study also did not account for geopolitical and global economic events that may have impacted the relationship between FDI and NE.

Future research can address these limitations by incorporating more recent data to capture evolving relationships between FDI and NE. Examining micro-level factors and firm-level data can provide deeper insights into the mechanisms driving FDI and NE. Studying the impact of geopolitical events and global economic crises on FDI and NE can offer valuable insights for policymakers. Case studies in specific countries or regions can provide context-specific insights into the relationship between FDI and NE. Additionally, exploring the role of technological advancements and digitalisation in influencing FDI and NE could be a relevant area of future research. By addressing these areas, future research can contribute to a more comprehensive understanding of the complex relationship between FDI and NE and inform more effective policy strategies for sustainable economic growth.”

---

## [Decision Letter · Decision Letter 1]

29 Aug 2023

From Short to Long Term: Dynamic Analysis of FDI and Net Export in Global Regions

PONE-D-23-12176R1

Dear Dr. Jayathilaka,

We’re pleased to inform you that your manuscript has been judged scientifically suitable for publication and will be formally accepted for publication once it meets all outstanding technical requirements.

Kind regards,

Fiza Qureshi, PhD

Academic Editor

PLOS ONE

Additional Editor Comments (optional):

Reviewers' comments:

Reviewer's Responses to Questions

**Comments to the Author**

1. If the authors have adequately addressed your comments raised in a previous round of review and you feel that this manuscript is now acceptable for publication, you may indicate that here to bypass the “Comments to the Author” section, enter your conflict of interest statement in the “Confidential to Editor” section, and submit your "Accept" recommendation.

Reviewer #1: All comments have been addressed

Reviewer #2: All comments have been addressed

2. Is the manuscript technically sound, and do the data support the conclusions?

Reviewer #1: Yes

Reviewer #2: Yes

3. Has the statistical analysis been performed appropriately and rigorously? 

Reviewer #1: Yes

Reviewer #2: Yes

4. Have the authors made all data underlying the findings in their manuscript fully available?

Reviewer #1: Yes

Reviewer #2: Yes

5. Is the manuscript presented in an intelligible fashion and written in standard English?

Reviewer #1: Yes

Reviewer #2: Yes

6. Review Comments to the Author

Reviewer #1: (No Response)

Reviewer #2: current version read well and suitable for journal audience

all section and previously raised issues been meet satisfactory

from the introductory to the concluding section

7. PLOS authors have the option to publish the peer review history of their article (what does this mean?). If published, this will include your full peer review and any attached files.

Reviewer #1: No

Reviewer #2: **Yes: **Festus Victor Bekun

---

## [Editor Report · Acceptance letter]

31 Aug 2023

PONE-D-23-12176R1 

From Short to Long Term: Dynamic Analysis of FDI and Net Export in Global Regions 

Dear Dr. Jayathilaka:

I'm pleased to inform you that your manuscript has been deemed suitable for publication in PLOS ONE. Congratulations! Your manuscript is now with our production department. 

Kind regards, 

on behalf of

Dr. Fiza Qureshi 

Academic Editor

PLOS ONE